# *Echis carinatus* snake venom metalloprotease-induced toxicities in mice: Therapeutic intervention by a repurposed drug, Tetraethyl thiuram disulfide (Disulfiram)

Gotravalli V. Rudresha[1], Amog P. Urs[2], Vaddarahally N. Manjuprasanna[1], Mallanayakanakatte D. Milan Gowda[1], Krishnegowda Jayachandra[1], Rajesh Rajaiah[3]*, Bannikuppe S. Vishwanath[1,3]*

1 Department of Studies in Biochemistry, University of Mysore, Manasagangotri, Mysore, Karnataka, India,
2 The Ohio State University Comprehensive Cancer Center, Columbus, Ohio, United States of America,
3 Department of Studies in Molecular Biology, University of Mysore, Manasagangotri, Mysore, Karnataka, India

* rajesh.r@biochemistry.uni-mysore.ac.in (RR); vishmy@biochemistry.uni-mysore.ac.in (BSV)

## Abstract

*Echis carinatus* (EC) is known as saw-scaled viper and it is endemic to the Indian subcontinent. Envenoming by EC represents a major cause of snakebite mortality and morbidity in the Indian subcontinent. Zinc ($Zn^{++}$) dependent snake venom metalloproteases (SVMPs) present in *Echis carinatus* venom (ECV) is well known to cause systemic hemorrhage and coagulopathy in experimental animals. An earlier report has shown that ECV activates neutrophils and releases neutrophil extracellular traps (NETs) that blocks blood vessels leading to severe tissue necrosis. However, the direct involvement of SVMPs in the release of NETs is not clear. Here, we investigated the direct involvement of EC SVMPs in observed pathological symptoms in a preclinical setup using specific $Zn^{++}$ metal chelator, Tetraethyl thiuram disulfide (TTD)/disulfiram. TTD potently antagonizes the activity of SVMPs-mediated ECM protein degradation in vitro and skin hemorrhage in mice. In addition, TTD protected mice from ECV-induced footpad tissue necrosis by reduced expression of citrullinated H3 (citH3) and myeloperoxidase (MPO) in footpad tissue. TTD also neutralized ECV-induced systemic hemorrhage and conferred protection against lethality in mice. Moreover, TTD inhibited ECV-induced NETosis in human neutrophils and decreased the expression of peptidyl arginine deiminase (PAD) 4, citH3, MPO, and p-ERK. Further, we demonstrated that ECV-induced NETosis and tissue necrosis are mediated via PAR-1-ERK axis. Overall, our results provide an insight into SVMPs-induced toxicities and the promising protective efficacy of TTD can be extrapolated to treat severe tissue necrosis complementing anti-snake venom (ASV).

**Data Availability Statement:** All relevant data are within the manuscript and its Supporting information files.

**Funding:** The study was funded by the University Grants Commission of India, a statutory body set up by the Government of India under Ministry of Education (MRP-MAJOR-BIOC-2013-12157 to BSV) and Science and Engineering Research Board, a statutory body under the Department of Science and Technology, Government of India (EEQ/2017/000737 to RR). Funders did not play any role in the study design, data collection and analysis, decision to publish, or preparation of the manuscript.

**Competing interests:** The authors have declared that no competing interests exist.

## Author summary

India has highest incidence of deaths due to snakebite in the world. *Echis carinatus* (EC) is known as saw-scaled viper and its bite causes major mortality and morbidity in the Indian subcontinent. The abundant presence of zinc ($Zn^{++}$) metalloproteases in *Echis carinatus* venom (ECV) is responsible for local tissue necrosis. An earlier report has shown that ECV activates neutrophils and leads to NETosis that blocks blood vessels leading to tissue necrosis. However, the toxin in ECV responsible for NETosis has not addressed. Here we investigated the *Echis carinatus* venom metalloproteases (ECVMPs) are responsible for NETosis and its associated tissue necrosis using $Zn^{++}$ specific chelator, Tetraethyl thiuram disulfide (TTD). TTD inhibited ECVMPs-induced skin hemorrhage and footpad tissue necrosis by reduced expression of citrullinated H3 (citH3) and myeloperoxidase (MPO) in mice footpad tissue. TTD also neutralized ECV-induced systemic hemorrhage and conferred protection against lethality in mice. Moreover, TTD inhibited ECV-induced NETosis in human neutrophils and decreased the expression of p-ERK and NETosis markers. Further, we demonstrated that ECV-induced NETosis and tissue necrosis is mediated via PAR-1-ERK axis. Overall, our results provide an insight into SVMPs-induced toxicities and the promising protective efficacy of TTD can be extrapolated to treat severe tissue necrosis complementing anti-snake venom (ASV).

## Introduction

According to the World Health Organization, snakebite is a global public health problem and a neglected tropical disease [1,2]. Snakebites are often associated with severe local manifestations including inflammation, hemorrhage, blistering, skin damage, coagulopathy and progressive tissue necrosis at the bitten site [3,4]. These local manifestations are nevertheless pathological condition, caused by a mixture of toxins rather than a single toxin present in the venom [4]. Unfortunately, treatment of snake envenomation-induced progressive tissue necrosis persists even after anti-snake venom (ASV) administration [5,6]. Hence, treating progressive tissue necrosis is still a challenging issue for the existing strategies of snakebite management. In addition, studies on progressive tissue necrosis induced by viper bite have clearly revealed the direct involvement of metzincin family matrix-degrading snake venom metalloproteases (SVMPs) [4,7,8] and hyaluronidases (SVHYs) [9,10].

*E. carinatus* (EC) also known as a saw-scaled viper is the only *Echis* species found in India [11]. *Echis* bite is responsible for 10–20% mortality and causes severe local manifestations, due to rich in zinc ($Zn^{++}$) dependent SVMPs [12–14]. In addition, ECV activates neutrophils and releases their nuclear and granular contents to the extracellular space leading to neutrophil extracellular traps (NETs); this process is described as NETosis [15]. A previous study showed that NETs released from neutrophils entrap the venom toxins and results in blockage of blood vessels leading to tissue necrosis [15]. However, the venom toxin/s responsible for NETs formation and cellular mechanism involved is unclear.

Several studies have demonstrated that SVMPs of ECV are known to contribute to the pathophysiology by causing degradation of extracellular matrix (ECM) proteins resulting in hemorrhage at the site of injection [4,8,16]. Multiple scientific reports have demonstrated the direct involvement of SVMPs in disrupting the tissue architecture by degrading ECM proteins [7,17,18]. Hemorrhagic SVMPs act at the basement membrane and disrupt the capillary wall that results in extravasation [7,17,19]. Further experimental evidence suggests that the onset of micro-vessel damage is mediated by the degradation of type IV collagen by the action of

SVMPs [7,19]. SVMPs have resemblance in catalytic site architecture and structural domains with metzincin family proteases such as MMPs and ADAMs [20]. A few reports showed the activation of MAPKs by MMPs via protease-activated receptor (PAR)-1 [21]. Since SVMPs are catalytically related to MMPs, we hypothesized that EC SVMPs-induce NETosis and intracellular signaling cascade via PAR-1. Here, we have demonstrated that EC SVMPs-induced NETosis is mediated via PAR-1-ERK signaling axis, responsible for severe tissue necrosis.

Previously, we have shown the neutralizing abilities of $Zn^{++}$ specific chelators against the snake venom-induced progressive tissue damage [22]. Very recently, Albulescu et al. demonstrated the therapeutic intervention of repurposed drug, 2, 3-dimercapto-1-propanesulfonic acid for hemotoxic snakebite [23]. Chelating agents are essential in restoring the physiological levels of MMPs, as their dysregulated activity reflects in debilitating conditions such as cancer and arthritis [24]. Many pharmacologically approved chelating agents have been extensively studied for inhibition of SVMPs [25,26]. Those molecules are failed to reach the clinical trial, because of their non-specific chelation property [27]. Consequently, a high affinity membrane permeable specific $Zn^{++}$ chelator, Antabuse drug, Tetraethyl thiuram disulfide (TTD)/disulfiram repurposed as therapeutic for ECV-induced toxicities in preclinical setup and compared with $PLA_2$ and hyaluronidase inhibitors, aristolochic acid (AA) and silymarin (SLN), respectively.

## Materials and methods

### Ethics statement

Adult Swiss albino mice (6 to 8-week-old female) weighing 20–25 g were obtained from the Central Animal House Facility, Department of Studies in Zoology, University of Mysore, Mysuru, India. The animal experiments were approved by the Institutional Animal Ethical Committee, University of Mysore, Mysuru, India (Approval number: UOM/IAEC/20/2016). During all experiments, animal care and handling were in accordance with the guidelines of the Committee for the Purpose of Control and Supervision of Experiments on Animals (CPCSEA).

Human blood was drawn from the antecubital veins of healthy adult volunteers who were provided with written informed consent. All the experiments were approved by the Institutional Human Ethical Committee, University of Mysore, Mysuru, India (Approval number: IHEC-UOM No. 120 Ph.D/2015-16), and conducted in accordance with the ethical guidelines.

### Venom

Lyophilized powder of *Echis carinatus* venom (ECV) was purchased from Irula Snake-Catchers Co-operative Society Ltd., (Chennai, India). The required amount of venom was re-dissolved in PBS and centrifuged at 9000 g for 10 min to remove debris. The protein content of venom was determined according to the method of Lowry et al. [28].

### Chemicals and reagents

Tetraethyl thiuram (TTD), aristolochic acid (AA), silymarin (SLN), phorbol 12-myristate 13-acetate (PMA), porcine skin gelatin, collagen-I/IV, laminin, fibronectin, Hoechst stain, DNase 1, ficoll-paque, dextran and phosphatase inhibitor cocktail were obtained from Sigma-Aldrich (Bangalore, India). BSA, ethanol, dimethyl sulfoxide (DMSO; HPLC grade), Tween-20 and Hank's balanced salt solution (HBSS) were purchased from HiMediaLaboratories, Pvt. Ltd. (Mumbai, India). SCH79797 (PAR-1 antagonist) and GB-83 (PAR-2 antagonist) were purchased from Cayman Chemicals (Michigan, USA). U0126 (MEK 1/2 inhibitor), antibodies

against p-ERK, β-actin and cell lysis buffer were purchased from Cell Signaling Technology (Massachusetts, USA). HRP tagged anti-rabbit IgG and anti-mouse IgG were procured from Jackson ImmunoResearch (Philadelphia, USA). The rabbit polyclonal anti-citH3, rabbit polyclonal anti-H3, mouse monoclonal anti-myeloperoxidase (anti-MPO) and anti-PAD4 were obtained from Abcam (Cambridge, UK). All other chemicals and reagents used in this study are analytical grade.

## PLA$_2$ activity

ECV PLA$_2$ activity was performed according to the method of Patriarca et al. with some modifications [29]. *E. coli* was labeled with $^{14}$C-oleate, autoclaved and used to measure PLA$_2$ activity. ECV (0–50 μg) was added into a total reaction volume of 350 μl containing 5 mM CaCl$_2$, 100 mM Tris-HCl buffer (pH 7.4) and $^{14}$C-oleate labeled *E. coli* cells ($3.18 \times 10^9$) (corresponds to 10,000 cpm or 60 nmol lipid phosphorus) and incubated at 37°C for 60 min. The reaction was terminated by adding 100 μl of 2N HCl and 100 μl of fatty acid free BSA (100 mg/ml). The tubes were vortexed, centrifuged at 20,000 g for 10 min and aliquot (140 μl) of supernatant was mixed with scintillation cocktail. The enzyme activity was determined by quantifying the free $^{14}$C-oleate released using Packard scintillation analyzer and expressed as nmols of free fatty acid released/min/mg of protein at 37°C. For inhibition studies, similar reactions were carried out after pre-incubating 50 μg ECV with various concentrations of AA, SLN and TTD for 5 min at 37°C. Inhibition was expressed as a percentage.

## Hyaluronidase activity

Hyaluronidase activity of ECV was assayed according to the method of Reissig et al. with some modifications [30]. The reaction mixture (350 μl in 0.1 M sodium acetate buffer pH 5.5 with 0.15 M NaCl) containing ECV (0–100 μg) incubated separately with HA (50 μg) at 37°C for 2½ h. After incubation, the reaction mixture was heated in a water bath for 5 min to stop the reaction and cooled to room temperature. Sodium tetraborate (50 μl; 0.8 M; pH 9.2) buffer was added followed by heating in a boiling water bath for 3 min. After cooling to room temperature, 1.5 ml of coloring reagent p-DMAB (1% in 9,1 ratio of glacial acetic acid and HCl) was added and incubated for 20 min at 37°C and centrifuged at 1500 *g* for 10 min to remove turbidity, the absorbance of clear supernatant was measured at 585 nm. Activity was expressed as μmol of NAG released/min/mg protein. For inhibition studies, hyaluronidase activity was determined after pre-incubating 100 μg ECV with various concentrations of AA, SLN and TTD for 5 min at 37°C. Inhibition was expressed as a percentage.

## Caseinolytic activity

Proteolytic activity of ECV was assayed according to the method of Murata et al. with suitable modifications [31]. Fat free casein 0.4 ml (2%; 0.2 M Tris-HCl buffer; pH 8.5) was incubated with ECV (0–25 μg) and final volume make up to 1 ml with 0.2 M Tris-HCl (pH 8.5), incubated at 37°C for 2½ h. The reaction was stopped by adding 1.5 ml of 0.44 M TCA and allowed to stand for 30 min. The mixture was centrifuged at 1,500 g for 15 min and 1.0 ml supernatant was mixed with 2.5 ml of 0.4 M sodium carbonate and 0.5 ml of 1:2 diluted Folin-Ciocalteu reagents. The colour developed was read at 660 nm. One unit of enzyme activity was defined as the amount of enzyme required to increase an absorbance of 0.01 at 660 nm/h at 37°C. For inhibition studies, similar reactions were performed after pre-incubating 25 μg of venom with various concentrations of AA, SLN and TTD for 5 min at 37°C. The proteolytic activity of ECV in the absence of inhibitors was considered as 100%. Inhibition was expressed as a percentage.

## Gelatinolytic activity

The gelatinolytic activity was performed by substrate gel assay as described by Heussen and Dowdle, with some modifications [32]. ECV, 5 μg was loaded onto a 10% SDS polyacrylamide gel (SDS-PAGE) impregnated with 0.08% of gelatinand electrophoresis was carried out under non-reducing condition at a 100 V for 2 h. After electrophoresis, SDS was removed by incubating gel with 2.5% Triton X-100 for 1 h, followed by an extensive wash with distilled water. The gel was incubated overnight at 37°C in incubation buffer, 50 mM Tris-HCl, pH 7.6 containing 0.9% NaCl, 10 mM $CaCl_2$, 10 mM $ZnCl_2$. The gel was stained with Coomassie brilliant blue-G250 (CBB-G250) and a clear zone indicates the gelatinolytic activity of ECV. For inhibition studies ECV was pre-incubated with different concentrations of TTD (1, 5, 10 and 20 mM), AA (10 and 20 mM) and SLN (10 and 20 mM) for 5 min at 37°C and assay was performed as described above.

## ECM protein hydrolyzing activity

ECM protein hydrolyzing activity was performed according to the method of Baramova et al. with slight modifications [33]. ECM proteins, collagen type-I/IV, laminin and fibronectin (50 μg each) were incubated with 5 μg of ECV, separately in a total reaction volume of 40 μl with Tris-HCl buffer (10 mM; pH 7.6) at 37°C for 3 h. The reaction was terminated by adding 20 μl of reducing sample buffer (4% SDS, 6% β-mercaptoethanol and 1 M urea) and boiled for 5 min. An aliquot of 40 μl of this sample was loaded onto 7.5% SDS-PAGE and electrophoresis was carried out at 100 V for 2 h. After electrophoresis the cleavage pattern of ECM proteins was visualized by staining with CBB-G250. For inhibition studies, similar experiments were carried out by pre-incubating ECV with different concentrations of TTD (1, 5, 10 and 20 mM), AA (10 and 20 mM) and SLN (10 and 20 mM) for 5 min at 37°C and electrophoresed as described above.

## ECV-induced skin hemorrhage in mice

Hemorrhagic activity was performed as described by Kondo et al. with suitable modifications [34]. Mice were injected (n = 3; i.d.) with 5 μg of ECV and control mice received saline. After 2½ h, mice were sacrificed using pentobarbitone (30 mg/kg; i.p.) and the inner dorsal surface of the skin was surgically removed and photographed, and the hemorrhagic spot was quantitated using a graph sheet. For neutralization studies, 5 μg of ECV was pre-incubated with various doses of TTD (5, 10 and 20 mM) for 5 min at 37°C and administered to the mice skin (i. d.). For challenge studies, various doses of TTD (13.25, 26.5 and 53 μg), AA (20 and 40 μg) and SLN (15 and 30 μg) were injected 15 min post ECV injection. Neutralization of hemorrhagic activity was measured in terms of decreased area of the hemorrhagic spot in comparison to ECV-induced hemorrhagic area.

## ECV-induced mice footpad tissue necrosis

ECV-induced mice footpad tissue necrosis was performed as described by Rudresha et al. with suitable modifications [35]. Mice were anesthetized using pentobarbitone (30 mg/kg; i.p.) and ECV ($LD_{50}$; 2.21 mg/kg body weight) was administered to the mice footpad (n = 5; Intraplantar injection). The time for onset of footpad injuries was recorded for each mouse. The severity of the injury was visually judged and scored based on a 5-point scale; 0-no injury, 1-edema with mild hemorrhage, 2-edema with severe hemorrhage causing discoloration of the footpad, 3-edema with severe hemorrhage and necrosis, 4-severe hemorrhage and necrotized footpad, 5-necrotized little toe detached from limb. The footpad injury observations were recorded

every day for 8 days after venom injection. To evaluate the protection of ECV-induced tissue necrosis, TTD (2.15 mg/kg) was pre-incubated with ECV (2.21 mg/kg) for 5 min at 37˚C and administered to the mice footpad. For challenge studies, TTD (2.15 mg/kg) was administered to the venom injection site 30 min post venom injection. Previously, Katkar et al. showed that DNase 1 administration accelerates the healing of ECV-induced chronic wounds by degrading the deposited NETs. Hence, DNase 1 (100 U) was used as a positive control [15].

To assess the effect of SCH79797 (PAR-1 antagonist) on ECV-induced footpad tissue necrosis, SCH79797 (1.5 mg/kg) was administered to the mice footpad 15 min before to the ECV (½LD$_{50}$; 1.10 mg/kg; n = 3) injection. The onset of footpad injuries was recorded as described above.

## Histopathological studies

Mice were sacrificed using pentobarbitone (30 mg/kg; i.p.) and footpad tissues were excised and fixed for 24 h in Bouin's fixative (picric acid: formaldehyde: glacial acetic acid, 30:10:2 v/ v), dehydrated with increasing concentrations of ethanol and embedded in paraffin. The tissues were sectioned into 4 μm thick using a microtome (R. Jung AG, Germany). Tissue sections were processed and stained with hematoxylin and eosin as described by Svensson et al. with suitable modifications [36] and photographed using Axio Imager A2 microscope with LED—Zeiss (Oberkochen, Germany).

## ECV-induced mortality and systemic hemorrhage in mice

Lethal toxicity of ECV was determined according to the method of Meier and Theakston, with some modifications [37]. To determine the anti-venom potential of TTD in preventive regimen we followed our previous lethality study [38]. Briefly, ECV (1½ LD$_{50}$; 3.31 mg/kg; n = 5) was pre-incubated with TTD (2.15 mg/kg) or effective dose ASV (ED ASV), separately for 5 min at 37˚C and injected intra peritoneally (i.p.) to mice. For challenge studies, TTD (2.15 mg/ kg) or ED ASV (mg anti-venom per mg venom) was injected 30 min post of venom administration (i.p.). The signs of toxicities were observed up to 24 h and survival time was recorded. For systemic hemorrhage and bleeding studies, mice received ECV (LD$_{50}$; 2.21 mg/kg; n = 5; i. p.) and challenged with TTD (2.15 mg/kg; i.p.) or ED ASV, 30 min post ECV injection. After 3 h mice were sacrificed using pentobarbitone (30 mg/kg; i.p.) and peritoneal cavity was observed for symptoms and photographed.

## Isolation of neutrophils from human blood

Human neutrophils were isolated from healthy volunteers blood, according to the method of Halverson et al. with slight modifications [39]. The blood was collected and mixed with acid citrate dextrose in a 5:1 volumetric ratio, followed by dextran sedimentation and hypotonic lysis to remove red blood cells. The cell pellet was suspended in 2 ml of PBS and subjected to density gradient centrifugation at 210 g using ficoll-paque for 30 min at 4˚C. The neutrophils settled at the bottom as a cell pellet were washed twice with PBS and centrifuged at 210 g and re-suspended in HBSS. The cells were counted using the Neubauer hemocytometer and the required cell density was adjusted using HBSS.

## Analysis of NET formation and its markers in human neutrophils

NETs formation was analyzed using Hoechst stain according to the method of Katkar et al. with suitable modifications [15]. Human neutrophils ($2\times10^5$/ml) were seeded on 13 mm round coverslips placed in 12 well plates in 500 μl of HBSS-1 and allowed cells to adhere to the

coverslips for 30 min at 37˚C in the presence of 5% $CO_2$. Then, the cells were stimulated with different concentrations of ECV (5, 10, 25 and 50 μg/ml) for 2½ h. After incubation, cells were fixed with 4% paraformaldehyde for 30 min and stained with Hoechst stain (1:10,000) for 15 min. Cells were analyzed for NETs formation and images were acquired on a BA410 fluorescence microscope (Motic) attached to a DS-Qi2 monochrome CMOS sensor camera. The NETs percentage was calculated in 5 non-overlapping fields per coverslip. For inhibition studies, similar experiments were carried out by pre-incubating ECV (25 μg) with different concentrations of TTD (1, 5, 10 and 20 mM), AA (10 and 20 mM) and SLN (10 and 20 mM) for 5 min at 37˚C and NETs percentage was calculated as described above.

For the analysis of ECV-induced citH3, PAD4 and MPO expression, neutrophils were treated with ECV (25 μg) that was pre-treated with various concentrations of TTD (1, 5, 10 and 20 mM), AA (10 and 20 mM) and SLN (10 and 20 mM) at 37˚C for 5 min. After 2½ h, neutrophils were washed and lysed with cell lysis buffer containing PMSF and phosphatase inhibitor cocktail and stored at -20˚C overnight. Lysates were centrifuged at 9000 g for 10 min and tested for the expression of protein using Western blotting.

To test the effect of pharmacological inhibitors on ECV-induced NETs and its markers, neutrophils were pre-sensitized without or with U0126 (MEK 1/2 inhibitor), SCH79797 (PAR-1antagonist) and GB-83 (PAR-2 antagonist) for 15 min. Then, neutrophils were stimulated with 25 μg of ECV for 2½ h at 37˚C, and NETs were quantitated using Hoechst stain. Cell lysates were used to test the expression of citH3, PAD4, MPO and ERK activation using Western blotting.

## Western blotting

Mice were sacrificed using anesthesia pentobarbitone (30 mg/kg; i.p.) and ECV injected mice footpads were excised from respective days 1 to 8 and stored at -20˚C. The excised tissue samples were sonicated for 30 seconds on ice bath (five passes of 10 seconds) with a 3.0 mm probe sonicator in cold cell lysis buffer containing (20 mM Tris-HCl (pH 7.5), 150 mM NaCl, 1 mM $Na_2EDTA$, 1 mM EGTA, 1% Triton, 2.5 mM sodium pyrophosphate, 1 mM beta-glycerophosphate, 1 mM $Na_3VO_4$, 1 μg/ml leupeptin) (Cell Signaling Technology) and supplemented with PMSF and phosphatase inhibitor cocktail (Sigma Aldrich). Tissue homogenates were centrifuged at 9000 g for 10 min and the supernatant was used for protein quantification. Western blotting was performed as described by Rajaiah et al. with suitable modifications [40]. Briefly, an equal amount of proteins (25 μg) with Laemmli buffer was loaded on to SDS-PAGE and electrophoresis were carried out at 100 V for 2 h. After electrophoresis proteins were electroblotted onto polyvinylidene difluoride (PVDF) membranes at 4˚C for 90 min. After blocking for 2 h in TBST (20 mM Tris, pH 7.5 with 150 mM NaCl and 0.05% Tween-20) containing 5% BSA, membranes were probed against anti-citH3 (1:1000) or anti-PAD4 (1:2000) or anti-MPO (1:1000) or anti-p-ERK (1:1000) antibodies overnight at 4˚C according to the manufacturer's instructions (Abcam, UK and Massachusetts, USA). Membranes were washed extensively (4x) using TBST and incubated with HRP-conjugated anti-rabbit/mouse IgG (1:10,000) for 1 h at room temperature. The blots were developed with an enhanced chemiluminescence substrate for visualization (Alliance 2.7, Uvitec) and bands were quantitated using Image J software. H3 and β-actin were used as the loading control.

## Statistics

Data represented as the mean ± SEM. One-way ANOVA followed by Bonferroni post hoc test was used to analyze more than two groups using Graph Pad Prism version 5.03 (La Jolla, USA). The comparison between the groups was considered significant if $^*p < 0.05$.

## Results

### TTD inhibits ECV-induced proteolytic activity, ECM protein degradation and hemorrhage

We have tested the inhibitory efficacy of an Antabuse drug, TTD with chelating property on ECV-induced proteolytic activity, ECM protein degradation and hemorrhage in mice and compared with pharmacological inhibitors of $PLA_2$ (AA) and hyaluronidase (SLN). To begin with, the effect of TTD on ECV-induced proteolytic activity was demonstrated by casein and gelatin as substrate. TTD inhibited ECV-induced caseinolytic and gelatinolytic activity in a concentration-dependent manner and the $IC_{50}$ concentration of caseinolytic activity was found to be 3 mM (Fig 1A and 1B and S1C and S3A Figs). In addition, TTD also inhibited ECV-induced degradation of the ECM proteins, collagen, laminin and fibronectin in a

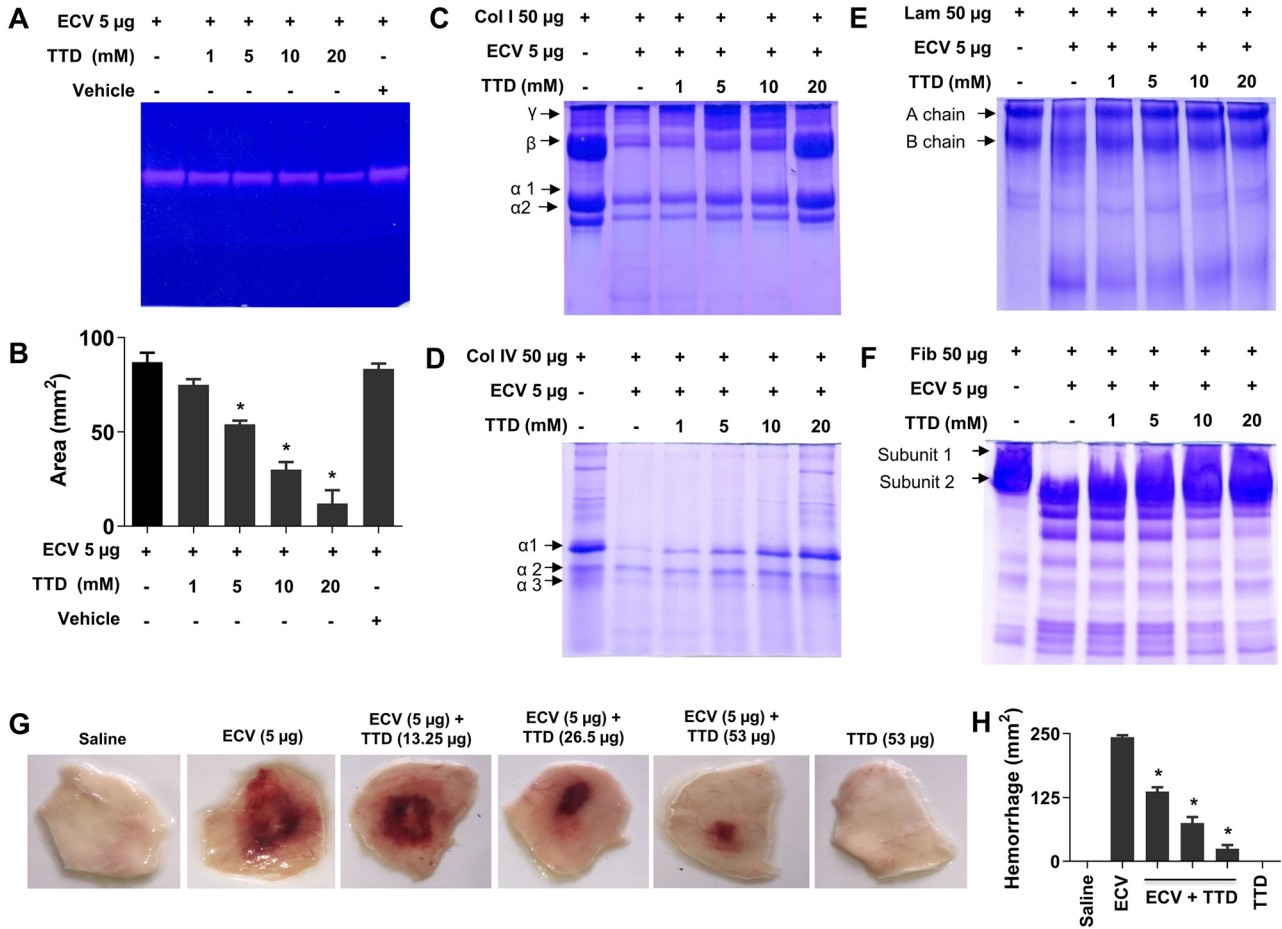

**Fig 1. Effect of TTD on ECV-induced ECM protein degradation in vitro and hemorrhage in mice.** ECV was pre-incubated without or with various concentrations of TTD at 37°C for 5 min and subjected for gelatinolytic, collagenolytic, laminin and fibronectin hydrolyzing activity. The gelatinolytic activity was performed by gelatin zymogram using 10% gel impregnated with 0.08% gelatin. Clear zones in the gel indicate the hydrolysis of gelatin by ECV **(A)**. The area of gelatinolytic activity was quantitated using a graph sheet represented as area ($mm^2$) **(B)**. The data represented as mean ± SEM. *p < 0.05, when compared ECV versus ECV + TTD. For collagen, laminin and fibronectin degradation analysis, the pre-incubated reaction mixture of ECV and TTD was incubated with 50 μg of collagen (Col) type I **(C)**, type IV **(D)**, laminin (Lam) **(E)** and fibronectin (Fib) **(F)** for 3 h at 37°C. The hydrolyzing pattern was analyzed using 7.5% SDS-PAGE and visualized by staining with CBB-G250. Data are representative of two independent experiments. For skin hemorrhage, mice were injected (n = 3; i.d.) with 5 μg of ECV followed by the injection of different concentrations of TTD post 30 min at the site of ECV injection. After 180 min, dorsal patches of mice skin were photographed **(G)** and the hemorrhagic area was measured using graph sheets represented as area ($mm^2$) **(H)**. Data are representative of two independent experiments.

concentration-dependent manner (Fig 1C–1F). Furthermore, TTD was tested for its action on ECV-induced hemorrhage in mice skin in both pre-incubation and challenging studies. TTD efficiently neutralized ECV-induced hemorrhagic activity in pre-incubation and 30 min post ECV injection (Fig 1G and 1H and S3B and S3C Fig). PLA$_2$ and hyaluronidase inhibitors, AA and SLN inhibited ECV-induced PLA$_2$ and hyaluronidase activities, respectively (S1A and S1B Fig). On the other hand, both AA and SLN failed to inhibit ECV-induced ECM protein degradation (S2A–S2D Fig) and hemorrhagic activity in mice (S2E and S2F Fig).

## TTD protects ECV-induced mice footpad tissue necrosis with decreased expression of citrullinated H3 (citH3)/myeloperoxidase (MPO) and histopathological changes

With promising results of in vitro inhibition of ECV-induced ECM proteins degradation and murine skin hemorrhage, TTD was tested for the neutralization of ECV-induced tissue necrosis using mice footpad model. ECV injection to mice footpad resulted in progressive tissue necrosis that leads to the detachment of little toe from limb between 6–8 days. TTD administration neutralizes ECV-induced tissue necrosis and prevented the loss of little toe and, was able to restore the normal footpad morphology both in pre-incubation and challenging studies (Fig 2A and 2B and S3D and S3E Fig). Moreover, recently Katkar et al. reported that infiltrated neutrophils to the site of venom injection release chromatin content to the extracellular space as NETs that is responsible for local tissue necrosis [15]. Furthermore, Katkar et al. and Rudresha et al. demonstrated that the intervention of DNase 1 and plant DNase at a right time protected ECV-induced tissue necrosis [15,41]. In addition, the excessive production of MPO and citH3 by the action of PAD4 has shown to be crucial for ECV-induced local tissue damage [15]. Similar to the previous study, ECV induced the expression of MPO and citH3, and it was efficiently inhibited by TTD (Fig 2C–2E). The inhibitory action of TTD on ECV-induced mice footpad necrosis and the expression of MPO and citH3 are more efficient and comparable with DNase 1 (Fig 2). In addition, the protective efficacy of TTD on ECV-induced footpad tissue necrosis was confirmed by histopathological studies using hematoxylin and eosin staining. Mice that received ECV alone showed extensive tissue damage. TTD treatment protects the ECV-induced histopathological changes (S4 Fig).

## TTD protects mice from ECV-induced lethality and neutralizes systemic hemorrhage

In addition to the induction of progressive tissue necrosis, ECV is lethal when injected at 3.31 mg/kg body weight (1½LD$_{50}$), and the average survival time is approximately $8 \pm 2$ h. Since TTD efficiently neutralized ECV-induced tissue necrosis and hemorrhage, its effect on ECV-induced mortality in mice was tested. TTD neutralized ECV-induced lethality and protected mice in both pre-incubation (100% survival—two independent experiments with 5 animals in each group) and challenge then treat (30 min post venom injection) (four of five animals survived—two independent experiments with 5 animals in each group) (Fig 3A and 3B). The protective effect of TTD was comparable to ED ASV (mg anti-venom per mg venom) both in pre-incubation and therapeutic regimens (Fig 3A and 3B). ECV is well-known for hemotoxic effect and its envenomation makes blood in-coagulable that leads to the systemic bleeding with disseminated intravascular coagulation [42]. In fact, ECV injection to mouse peritoneum caused severe bleeding and extravasation throughout the peritoneum (Fig 3C). As TTD protected mice from ECV-induced lethality, it neutralized ECV-induced bleeding in peritoneum even after 30 min post ECV injection and it was comparable with ED ASV as shown in Fig 3C. This indicates that TTD is a potential drug candidate that complements ASV during EC bite.

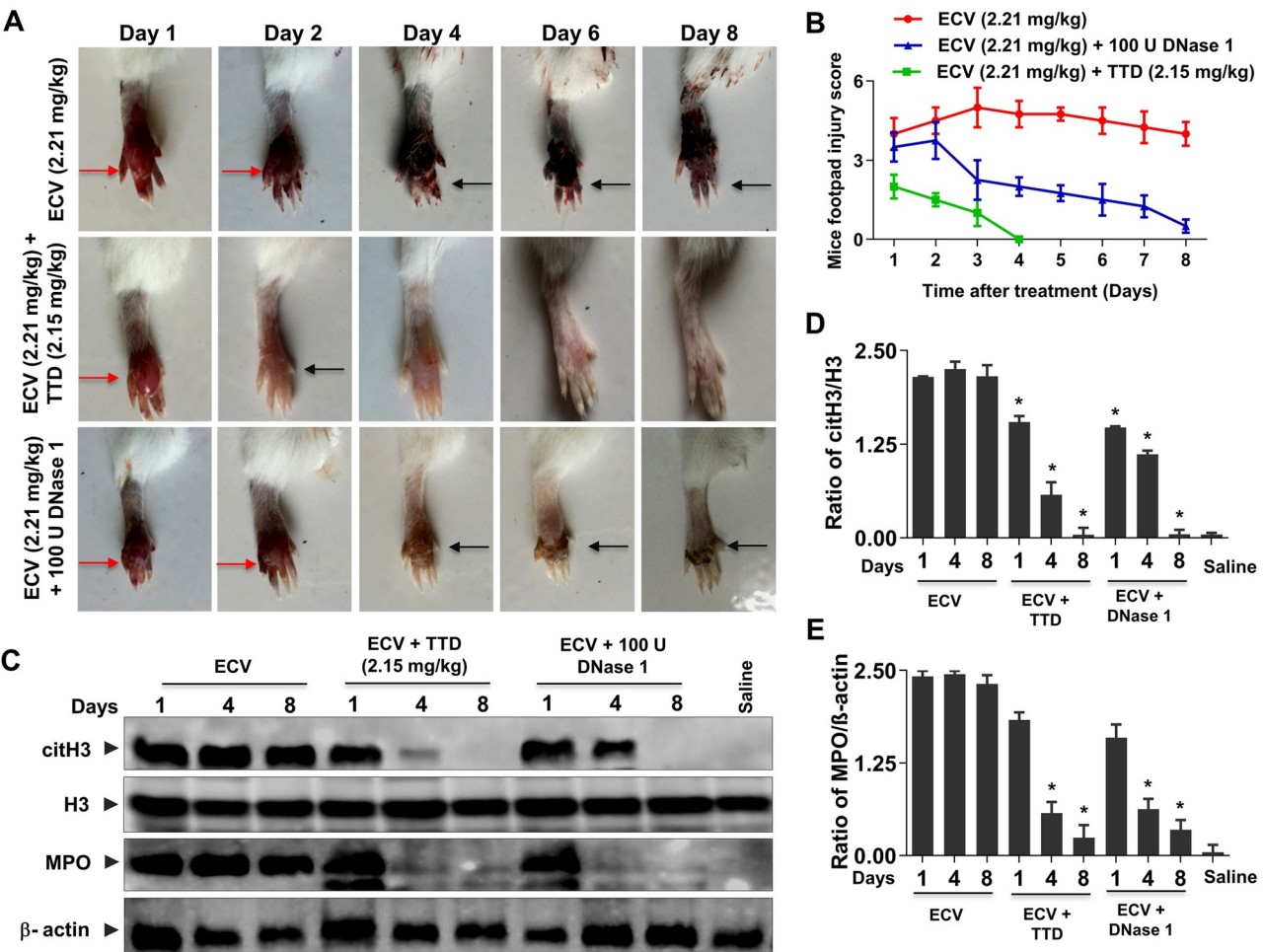

**Fig 2. Neutralization of ECV-induced mice footpad tissue necrosis by TTD.** Mice footpads were injected with ECV (LD$_{50}$; 2.21 mg/kg; n = 5). After 30 min, mice received either TTD or DNase 1 at the site of venom injection and footpads were photographed from day 1 to day 8 (**A**). Red arrow indicates edema and black arrow indicates tissue necrosis. ECV-induced footpad injury was measured manually on a scale of 1 to 5 (**B**). The level of ECV-induced citH3 and MPO in mouse footpad tissue in the absence or presence of either TTD or DNase 1 was analyzed by Western blotting (**C**) and quantitated using H3 and β-actin as a loading control for citH3 (**D**) and MPO (**E**), respectively. The data represented as mean ± SEM. *p < 0.05, when compared ECV versus ECV + TTD and ECV versus ECV + DNase 1.

## TTD inhibits ECV-induced NETs formation and activation of intracellular signaling in human neutrophils

Neutrophils are the first line innate immune cells recruited to sites of acute inflammation in response to chemotactic signals produced by injured tissue and tissue-resident macrophages [43,44]. During infection, neutrophils undergo degranulation and ultimately release chromatin as NETs that contribute to the killing of extracellular pathogens [45]. Previously, Setubal et al. demonstrated *Bothrops bilineatus* venom in the activation of neutrophils and the release of NETs [46]. Recently, Katkar et al. reported the discharged chromatin (NETs) upon ECV treatment is responsible for ECV-induced local tissue necrosis [15]. Similar to the previous reports, we observed ECV-induced chromatin discharge from human neutrophils in a concentration-dependent manner and it was effectively inhibited by TTD (Fig 4A and S5A Fig). On the other hand, the PLA$_2$ and hyaluronidase inhibitors, AA and SLN are failed to inhibit the

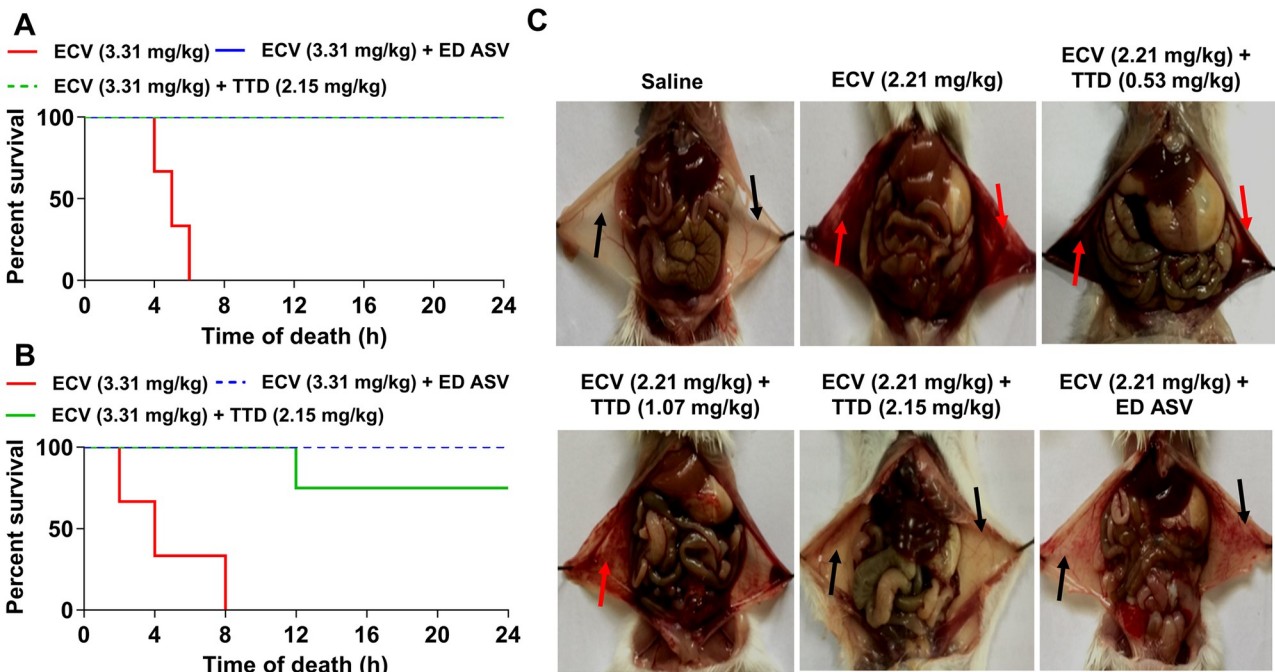

**Fig 3. Protection of mice against ECV-induced mortality and systemic hemorrhage by TTD.** A lethal dose of ECV ($1\frac{1}{2}LD_{50}$; 3.31 mg/kg) was pre-treated with TTD (2.15 mg/kg) or an effective dose of anti-snake venom (ED ASV) for 5 min at 37°C and injected (n = 5; i.p.) to mice. The time taken for mice mortality was recorded for 24 h and graph plotted as percent survival against the time of death (**A**). In the treatment model, mice received either TTD (2.15 mg/kg) or ED ASV, 30 min post ECV injection (i.p.) and the survival time was recorded for 24 h (**B**). For the neutralization of systemic hemorrhage, mice received (n = 5; i.p.) various concentrations of either TTD or ED ASV, 30 min post ECV ($LD_{50}$; 2.21 mg/kg; i.p.) injection. Mice were sacrificed after 2 h and peritonea were photographed (**C**). Red arrow indicates the hemorrhage in the peritoneum cavity and black arrow indicated reduced hemorrhage in the peritoneum. Data are representative of two independent experiments.

ECV-induced NETosis (S6A and S6B Fig). Moreover, ECV treated neutrophils showed increased expression of PAD4, citH3, and MPO and activation of ERK (Fig 4B). The importance of PAD4 in DNA de-condensation by citH3 and DNA expulsion in both mouse and human neutrophils is well documented [47]. Furthermore, TTD significantly reduced ECV-induced NETosis and decreased the expression of PAD4, citH3 and MPO as well as activation of ERK in neutrophils (Fig 4A and 4B). TTD is a chelating agent that is known to inhibit SVMPs; therefore, these data clearly suggest that SVMPs are directly involved in the activation of ERK and NETs formation.

## ECV-induced NETs formation and tissue necrosis via PAR-1-ERK mediated axis

It is well known that MMPs cleave PAR-1 at non-canonical sites, results in the activation of intracellular signaling cascade via MAPKs that leads to a diverse array of physiological functions [21,48]. Since MMPs and SVMPs are having structural homology in their catalytic site, we speculated that EC SVMPs activates ERK and NETs formation through PAR-1. To confirm whether ECV induces NETs formation via the PARs, we used PAR-1 and PAR-2 specific antagonists, SCH79797 and GB-83, respectively. SCH79797 is a selective antagonist of PAR-1 and it does not have any role in the inhibition of venom-induced toxicities by directly acting on ECV unlike TTD. SVMPs present in ECV instantaneously activate PAR-1 in the absence of

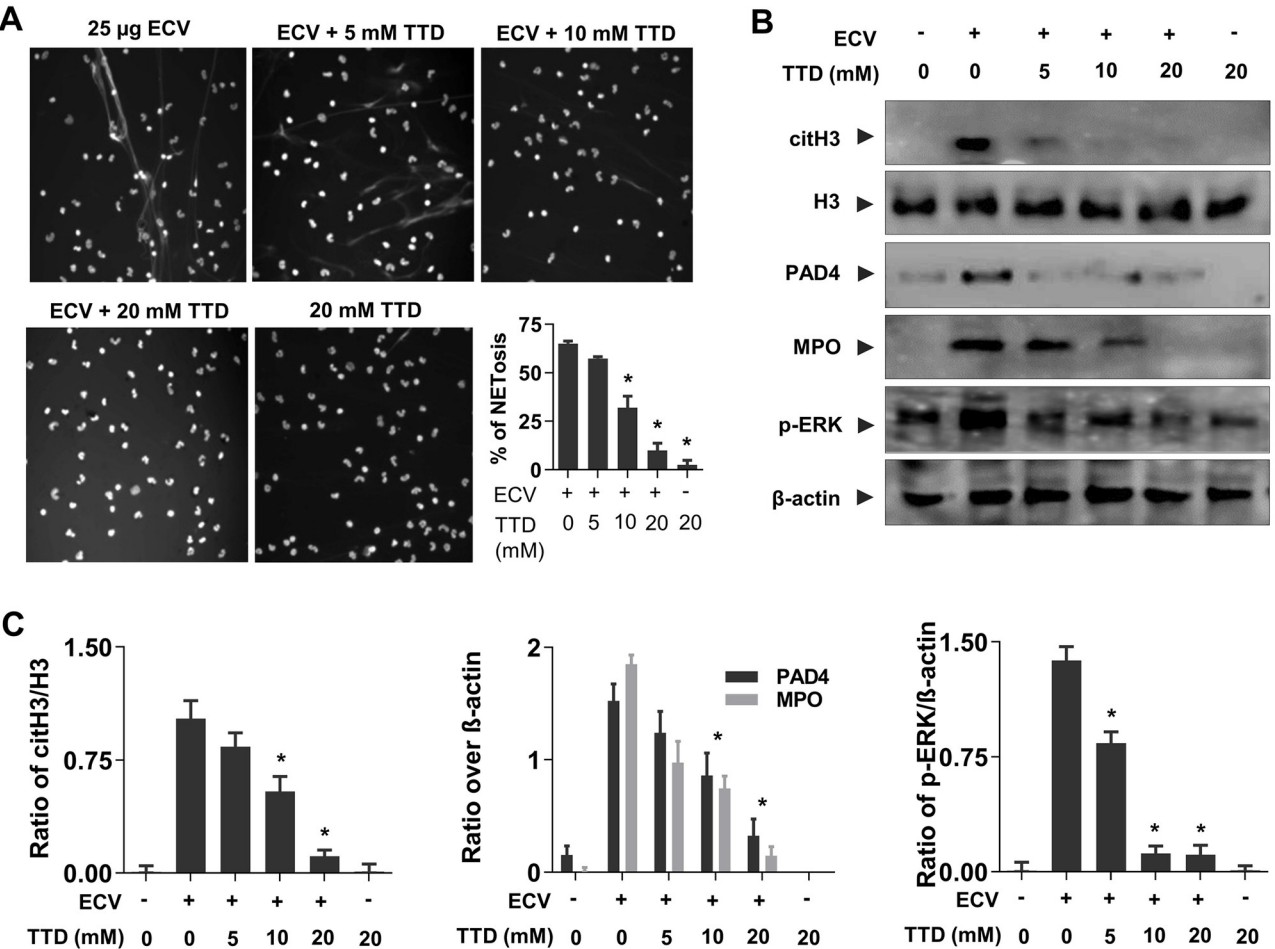

**Fig 4. Inhibition of ECV-induced NETs formation by TTD.** Human neutrophils were stimulated with ECV (25 μg) pre-incubated (5 min) without or with different concentrations of TTD for 180 min and NETs formation was observed and quantitated (**A**). ECV-induced citH3, PAD4 and MPO in neutrophil cell lysates were analyzed using Western blotting (**B**). Bands were quantitated using H3 as loading control for citH3 and β-actin as a loading control for MPO and PAD4 (**C**). The data represented as mean ± SEM. *p < 0.05, when compared ECV versus ECV + TTD.

TTD and it is hard to control SVMPs activated PAR-1 signaling once it is activated. Hence, we have reduced the venom dose and injected SCH79797 before ECV injection. With this, we made an effort to establish the mechanism of SVMPs in the activation of PAR-1 that may have direct/indirect role in ECV-induced toxicities. In fact, ECV activated NETs formation was inhibited in the presence of SCH79797 and not by GB-83, suggesting that ECV-induced NETosis is mediated via PAR-1 in human neutrophils (Fig 5A and 5B).

Further, ECV-induced expression of PAD4, CitH3 and activation of ERK was inhibited by SCH79797 (Fig 5C). On the other hand, MEK inhibitor, U0126 showed a partial effect on ECV-induced NETs formation and the expression of PAD4 and citH3 (Fig 5C). In support of *in vitro* results in human neutrophils, PAR-1 antagonist neutralized ECV-induced mice foot-pad tissue necrosis (Fig 6A and 6B). Overall, these data confirmed the involvement of EC SVMPs-induced tissue necrosis by inducing NETosis and activation of intracellular signaling via PAR-1 (Fig 7).

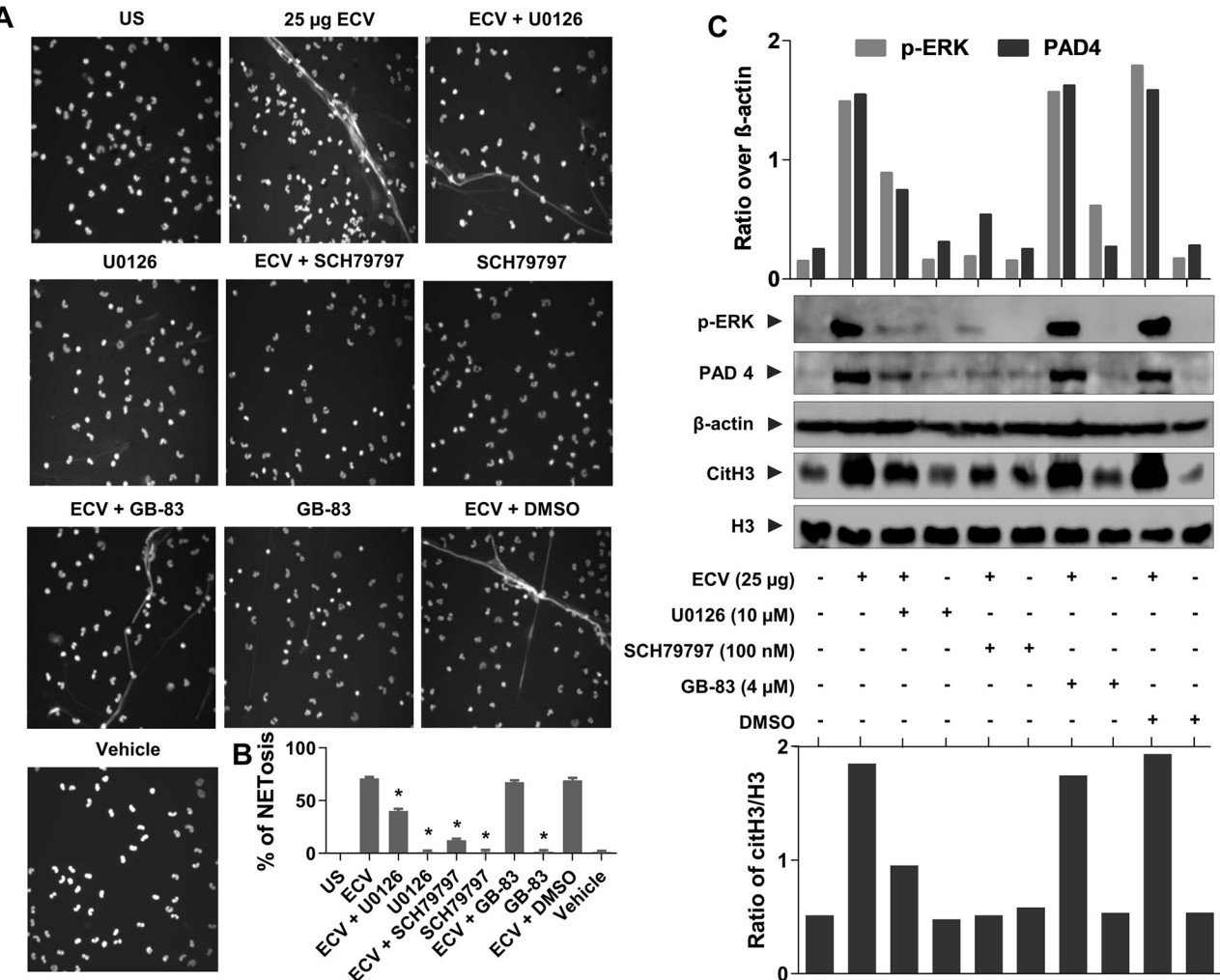

**Fig 5. Effect of selective antagonists of ERK and PARs on ECV-induced NETosis and tissue necrosis.** Neutrophils were pre-sensitized with selective antagonists of ERK (U0126), PAR-1 (SCH79797) and PAR-2 (GB-83) for 15 min, separately. Pre-sensitized neutrophils were stimulated with 25 μg of ECV for 180 min and cells were stained with Hoechst stain. Neutrophils were photographed under a microscope (**A**) and, NETs were quantitated and represented as percent NETosis (**B**). The data represented as mean ± SEM. *p < 0.05, when compared ECV versus ECV + antagonists. The whole cell lysates were analyzed for the phosphorylated ERK, expression of PAD4 and citH3 using Western blotting (**C**). The p-ERK and PAD4 bands were quantitated using β-actin as a loading control. The citH3 bands were quantitated using H3 as a loading control. Data are representative of two independent experiments.

## Discussion

Viper bites can induce progressive tissue necrosis that can result in permanent disability in the affected limb or digit [49]. Case reports on snakebite victims suggested that envenomation by hemotoxic venoms including *Echis carinatus* (EC) induces hematological complications, local pain, bleeding and edema at the bite site. Untreated *Echis* envenomation may involve multiple organs and the patient may suffer from, hematuria, renal failure, hemorrhage, anemia, hypotension and disseminated intravascular coagulation with capillary leak syndrome [50,51]. The $LD_{50}$ of EC envenomation is 6.65 mg/kg and an average bite may yield about 40 mg of venom [52–54]. Generally, envenomation by EC is associated with a mortality rate of 10–20% if there

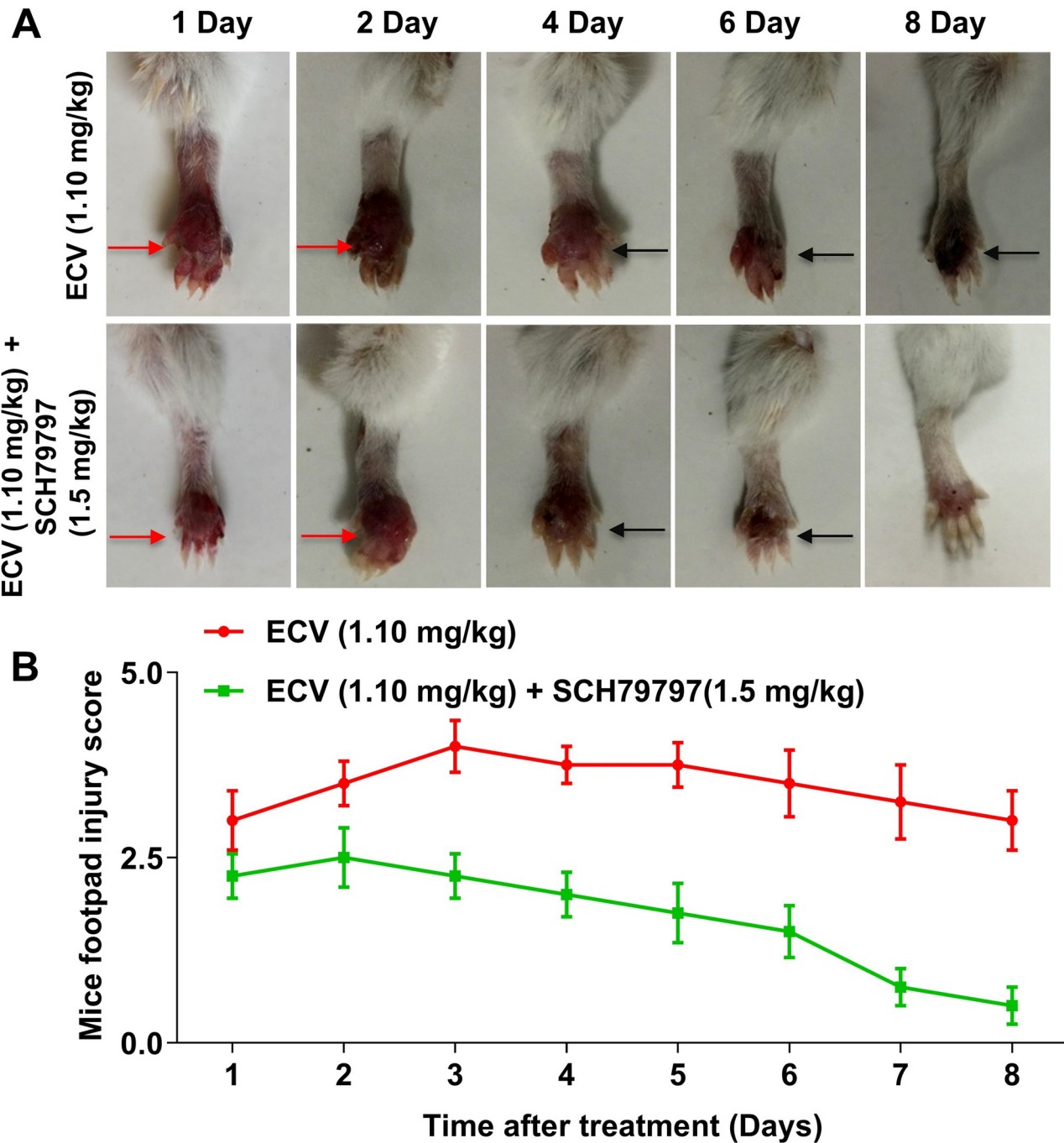

**Fig 6. Effect of the selective antagonist of PAR-1 on ECV-induced tissue necrosis.** Mice footpads (n = 5) were pre-treated without or with PAR-1 antagonist (SCH79797) for 15 min and followed by the injection of ECV (½LD$_{50}$; 1.10 mg/kg). Mice footpads were photographed from day 1 to day 8 (**A**) and tissue injury was measured manually on a scale of 1 to 5 (**B**). Red arrow indicates edema and black arrow indicates tissue necrosis. Data are representative of two independent experiments.

is no initiation of effective treatment early. The major cause of mortality is due to increased bleeding after envenomation including venom-induced consumption coagulopathy or disseminated intravascular coagulation due to the prothrombin/thrombin-like enzymes present in the snake venom [14].

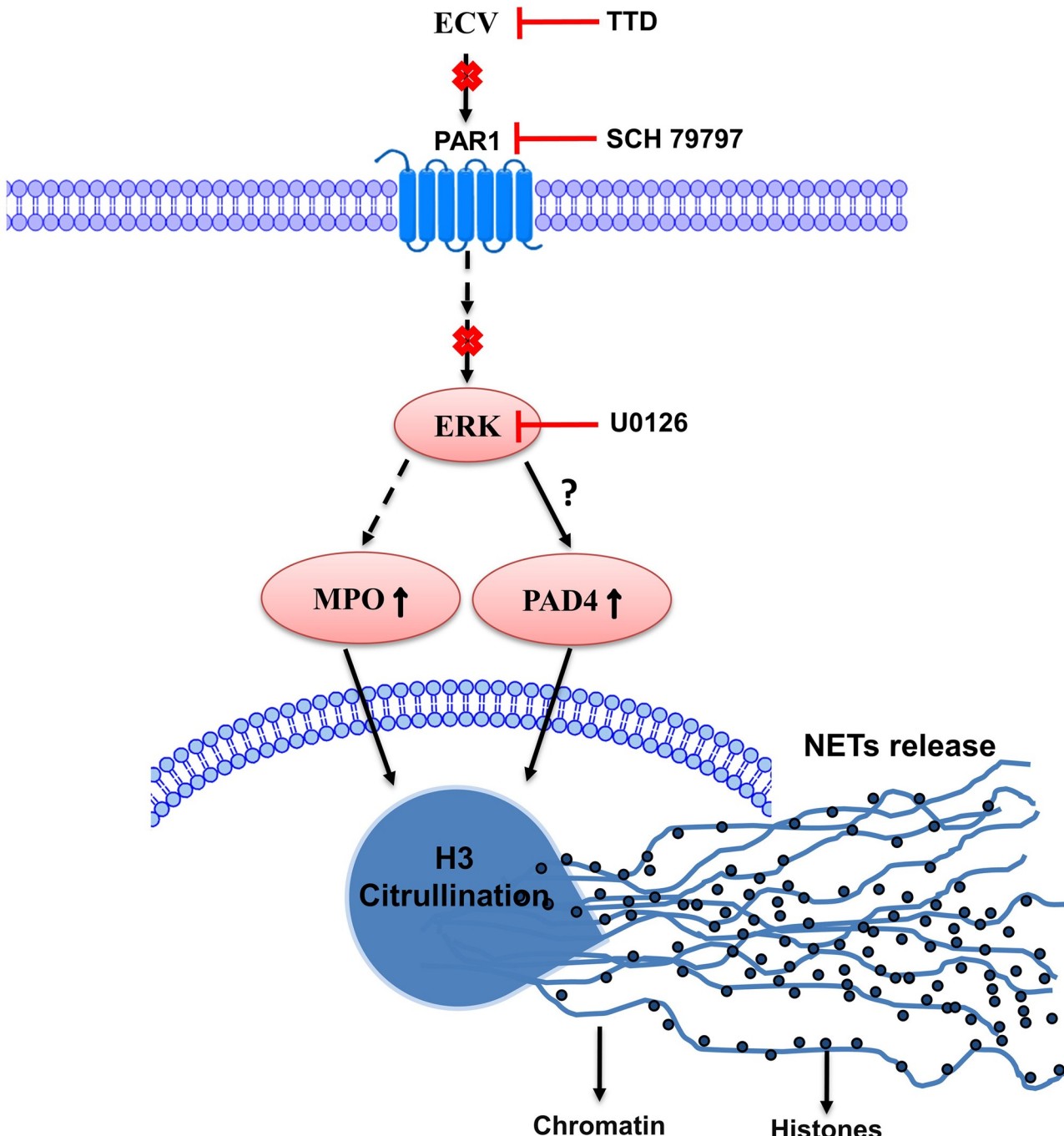

**Fig 7. Schematic representation of TTD and pharmacological inhibitors, site of action on ECV-induced toxicities.** ECV-induced PAR-1-mediated ERK activation might be responsible for increased expression of PAD4, histone citrullination and MPO expressions that are accountable for severe tissue necrosis. TTD and pharmacological inhibitors interfere in ECV-induced signaling/tissue necrosis by inhibiting NETosis and chromatin release.

SVMPs are one of the major toxins in most of the viper venoms including ECV and they primarily act on ECM components and are responsible for hemorrhagic activity [7,8,18,19,22]. The progressive tissue necrosis induced by viper bites mainly attributed to SVMPs, particularly P-III class metalloproteases [8]. In addition, SVMPs are hemotoxic in nature and interfere

with the hemostatic system in snakebite victims [55]. SVMPs are closely related to a disintegrin and metalloproteinases (ADAMs), thus they are also referred to as snake venom ADAMs [56,57]. SVMPs contains disintegrin-like (D), cysteine-rich (C), metalloproteinase and (M) domain, that harbors putative $Zn^{++}$ binding sequence and bearing the typical structural features of the metzincin family of MMPs [20,57–59]. Alike MMPs, $Zn^{++}$ on the M-domain of SVMPs plays a crucial role in the catalytic functions [20,57]. Hence, chelation of $Zn^{++}$ metal ion by specific $Zn^{++}$chelators rather than non-specific chelator is more effective in the management of local tissue necrosis induced by viper venoms [22]. Since SVMPs are directly responsible for ECV-induced toxicities, the inhibition of SVMPs by TTD would be beneficial to manage ECV-induced toxicities.

TTD was the first drug to treat chronic alcoholism and was approved by the FDA 1951 [60]. Since then, many studies have shown repurposing of TTD to treat diverse types of human malignant tumors including breast cancer, glioblastoma and pancreatic carcinoma [61,62]. TTD has also shown therapeutic potential in treating AIDS and it is found to be beneficial in treating Lyme disease in patients [63,64]. Very recently, the intervention of TTD in normalizing body weight in obese mice has been reported [65]. Besides, TTD has been shown to inhibit MMP-2 and MMP-9 activity by directly interacting with them via a $Zn^{++}$ chelating mechanism [66]. Several scientific reports suggested that, many small inhibitors or chelators of SVMPs such as batimastat, marimastat, N,N,N′,N′-tetrakis (2-pyridylmethyl) ethane-1,2-diamine and dimercaprol which targets the different classes of SVMPs-induced toxicities [67–71]. Some of these inhibitors were less effective as therapeutic regimen where the time lapse between venom and administration of inhibitors is prolonged [72]. Moreover, many small molecules and chelators were focused on interference in Viperinae snake venom-induced coagulopathy and local toxicities [69]. However, our findings highlight the efficacy of TTD in ECV-induced both local and systemic toxicities and, would be better repurposing to complement snakebite management. It has been demonstrated that 80–95% of an oral dose of TTD is absorbed from the gastrointestinal tract and it requires 1 to 2 h to peak serum concentration. TTD is rapidly distributed in adipose tissue, liver, spleen, adrenal gland and the brain. Prolonged administration of TTD is not known to induce tolerance and it is metabolized to diethyldithiocarbamate and mixed disulfides that are excreted via urine. The unabsorbed content of TTD is excreted in the feces [73,74]. TTD is known to cause hepatitis in 1 in 30,000 patients, which is sometimes fatal. There are rare reports of psychosis and confusional states and peripheral neuropathy and optic neuritis and, these effects were dose dependent. Moreover, TTD interacts with compounds that utilize the cytochrome P450 enzyme system [74,75]. Although the maximum recommended daily dose of TTD is 500 mg orally as an Antabuse [76], the long-term side effects of its use and dosage requirements are still unknown that requires extensive in vivo research before they can be fully supported as a complementary therapy for snakebite management.

Recently, Albulescu et al. showed that 2, 3-dimercapto-1-propanesulfonic acid, a derivative of dimercaprol effectively antagonizes the activity of $Zn^{++}$ dependent SVMPs in vitro and neutralized ECV in mice [23]. Previously, we have reported the inhibitory potential of $Zn^{++}$ specific chelating agents; N,N,N',N'-tetrakis (2-pyridylmethyl) ethane-1,2-diamine, diethylene triaminepenta acetic acid, TTD on ECV-induced toxicities [22]. In sight of these, we demonstrated that $Zn^{++}$ chelating agent, TTD an Antabuse drug can be likely repurposed as a therapeutic candidate in treating ECV-induced toxicities that are mediated by SVMPs. The proficient hydrolysis of the basement membrane by SVMPs surrounding the blood vessels leads to immediate events of hemorrhage at the site of venom injection [18,77]. The progression of hemorrhage resulting in localized myonecrosis is due to extensive degradation of structural proteins and severe inflammation [46,78]. Initially, TTD successfully inhibited ECV-induced degradation of ECM proteins in a concentration-dependent manner and also neutralizes the hemorrhage

induced by ECV upon challenging studies (Fig 1). On the other hand, AA and SLN inhibitors failed to inhibit the action of ECV-induced ECM protein degradation and hemorrhage. In support of the neutralization of hemorrhage, TTD treatment could efficiently protect mice footpad from ECV-induced necrosis (Fig 2). ECV-induced footpad necrosis is evident after day 4 of injection and necrotized little toe detached from limb between 6–8 days of ECV injection. This prompted us to carry out mice footpad necrosis experiments till 8 days after ECV injection. The successful neutralization of ECV-induced ECM proteins degradation and hemorrhage by TTD indicates that SVMPs are the main toxins responsible for ECV-induced toxicities. Further, EC SVMPs are also hemotoxic and interfere in hemostasis by hydrolyzing clotting factors that lead to persistent coagulopathy and death [79]. Most SVMPs are α and β fibrinogenases that act on fibrinogen and making them truncated, and non-functional [79]. A few scientific reports have shown that inhibitors of SVMPs effectively protect mice from viperid snake venom-induced lethality [22,23]. Similarly, TTD was effective in protecting mice from ECV-induced lethality and systemic hemorrhage (Fig 3). These data clearly indicate that TTD has a beneficial effect on neutralizing ECV-induced toxicities in mice.

Neutrophils are the first-line defense immune cells and efficiently arrest pathogens by NETosis at the site of infection [45,80]. Porto et al. demonstrated the infiltration of neutrophils at the site of viper venom injection [81]. However, the importance of NETosis in ECV-induced toxicities was not clear until Katkar et al. reported the critical role of NETosis in ECV-induced local tissue damage [15]. NETosis results in the blockage of blood vessels preventing venom from entering into the circulation. The accumulated venom-NETs complexes at the site of venom injection lead to the progressive tissue necrosis [15]. In addition, NETosis in non-healing wounds is noticeable by increased expression of PAD4, citH3 and MPO level [15,82]. However, the previous study did not explain in the context of the toxin that is responsible for ECV-induced NETosis and toxicities [15]. The inhibition of ECV-induced NETosis and reduced levels of PAD4, citH3 and MPO expression by TTD confirms the direct involvement of EC SVMPs in the induction of NETosis.

Nonetheless, the neutralized ECV-induced tissue necrosis and systemic hemorrhage by TTD correlated with the decreased ECV-induced NETosis. However, the mechanism of how ECV/SVMPs induce NETosis and toxicities is largely unknown. There are multiple scientific reports suggesting that the MMPs exert their effects by cleaving PARs and play an important role in vascular functions [21,48]. Moreover, MMPs bind and cleave the extracellular N-terminus of PAR-1 to release a tethered ligand and activate the intracellular G proteins across the membrane and initiate intracellular signaling cascade [21,83]. The inhibition of MMP-1 induced PAR-1 cleavage restricts the activation of MAPKs [84]. SVMPs belong to metzincin super-family and they are known to activate MAPKs signaling pathways in immune cells which results in elevated levels of pro-inflammatory mediators such as TNF-α, IL-1β and IL-6 leading to chronic inflammation [85]. Similarly, EC SVMPs mediates the phosphorylation of ERK in human neutrophils and it was completely inhibited by TTD (Fig 4). Similar to MMP-1, EC SVMPs might cleave PAR-1 at the non-canonical site and activate downstream MAPKs signaling. Finally, ECV-induced NETosis and tissue necrosis in experimental animals are effectively neutralized by PAR-1 antagonists (Figs 5 and 6). Overall, current findings indicate that direct involvement of PAR-1 and downstream MAPKs signaling cascade in EC SVMPs-induced toxicities in mice (Fig 7).

## Conclusion

There is an urgent need for effective snakebite treatments that can be administered in the remote areas where medical access is limited and also that can complement ASV. The current

findings suggested that TTD is a pharmacologically approved an Antabuse drug and that inhibits ECVMPs-induced NETosis in human neutrophils and footpad tissue necrosis in mice. Moreover, TTD also neutralized ECV-induced systemic hemorrhage and conferred protection against lethality in mice. Furthermore, we demonstrated that ECVMPs-induced NETosis and tissue necrosis is mediated via PAR-1-ERK axis. Overall, our results provide an insight into SVMPs-induced toxicities and the promising neutralizing potency of TTD can be exploited as first aid therapy, complementing ASV to treat snakebite-induced toxicities.

## Supporting information

**S1 Fig. Inhibition of ECV-induced enzymatic activities by specific inhibitors.** ECV was pre-incubated without or with various concentrations of AA/TTD/SLN at 37˚C for 5 min and subjected for PLA$_2$ **(A)**, hyaluronidase **(B)** and protease **(C)** activity. The inhibition was represented as % inhibition and venom alone considered as 100% activity. $^*$ $p < 0.05$, when compared ECV versus ECV + AA, ECV + SLN and ECV + TTD.
(TIF)

**S2 Fig. Effect of AA and SLN on ECV-induced ECM protein degradation and hemorrhage in mice.** ECV was pre-incubated without or with different concentrations of either AA **(A)** or SLN **(C)** at 37˚C for 5 min and subjected to gelatin zymogram as described in methods section. Clear zones in the gel indicate the hydrolysis of gelatin by ECV. Area of gelatinolytic activity was measured using graph sheet represented as area (mm$^2$) **(A and C)**. For collagen I (Col I), degradation, ECV was pre-incubated without or with increased concentrations of either AA **(B)** or SLN **(D)**. Pre-incubated reaction mixture of ECV and inhibitors were further incubated with 50 μg of collagen I for 3 h at 37˚C and cleavage pattern was analyzed using 7.5% SDS-PAGE and visualized by staining with CBB-G250. For skin hemorrhage, mice were injected (n = 3; i.d.) with 5 μg of ECV followed by two different concentrations of AA and SLN after 30 min venom injection. After 180 min, dorsal patches of mice skin were photographed **(E and F)**. Data are representative of two independent experiments.
(TIF)

**S3 Fig. Inhibition of ECV-induced protease activity and local toxicities by TTD.** Reaction mixture (1 ml) contained 0.4 ml of casein (2%) in 0.2 M Tris-HCl buffer pH 8.5 was incubated for 150 min at 37˚C with 25 μg of ECV and various concentrations of TTD (0–20 mM). The inhibition was represented as % inhibition and IC$_{50}$ (median inhibitory concentration) of the TTD was calculated **(A)**. For inhibition of skin hemorrhage, mice were injected (i.d.) with 5 μg of ECV that was pre-incubated with different concentrations of TTD (0–20 mM) at 37˚C for 5 min. After 180 min, dorsal patches of mice skin were photographed and IC$_{50}$ (median inhibitory concentration) of the TTD was calculated **(B and C)**. For inhibition of tissue necrosis, mice foot-pads were injected with ECV (LD$_{50}$; 2.21 mg/kg) pre-incubated with TTD (20 mM) at 37˚C for 5 min and footpads were photographed from day 1 to day 8 **(D)**. Red arrow indicates edema and black arrow indicates tissue necrosis. ECV-induced footpad injury was measured manually on a scale of 1 to 5 **(E)**. Data are representative of two independent experiments.
(TIF)

**S4 Fig. Histochemical staining of ECV-induced tissue necrosis in mice footpad and its inhibition by TTD.** Mice footpad was injected with ECV followed by TTD or DNase 1 injection (30 min post venom injection). Mice were euthanized and footpad tissues were processed for histological sections and analyzed for tissue damage by H & E staining. PBS injected mouse footpad serves as control.
(TIF)

**S5 Fig. ECV-induced NETosis and activation of intracellular signaling in neutrophils.** Human neutrophils were stimulated with ECV for 180 min and NETs formation was observed and quantitated (**A**). The whole cell lysates were analyzed for the phosphorylated ERK and NETosis markers using Western blotting. The p-ERK, MPO and PAD4 were quantitated using β-actin as a loading control and H3 as loading control for citH3. Data are representative of two independent experiments.
(TIF)

**S6 Fig. Effect of AA and SLN on ECV-induced NETs formation.** Human neutrophils were stimulated (180 min) with ECV pre-incubated with different concentrations of either AA or SLN for 5 min at 37°C and NETs formation was photographed under microscope (**A**) and quantitated as percent NETosis (**B**). The data represent the mean ± SD of three independent experiments. * $p < 0.05$, when compared ECV versus ECV + AA and ECV + SLN.
(TIF)

## Acknowledgments

Authors thank Central Animal Facility, University of Mysore, for providing animals. Authors thank Prof. Manjunath Kini R, Dr. Nanjaraj UrsA N, Dr. Vikram Joshi and Dr. Suvilesh K N for providing valuable suggestions during the project. Authors also thank -Dr. Sumanth M S and Dr. Abhilasha K V for their help during experiments and neutrophils isolation.

## Author Contributions

**Conceptualization:** Gotravalli V. Rudresha, Rajesh Rajaiah, Bannikuppe S. Vishwanath.

**Formal analysis:** Gotravalli V. Rudresha, Rajesh Rajaiah, Bannikuppe S. Vishwanath.

**Funding acquisition:** Rajesh Rajaiah, Bannikuppe S. Vishwanath.

**Investigation:** Gotravalli V. Rudresha, Amog P. Urs, Vaddarahally N. Manjuprasanna, Mallanayakanakatte D. Milan Gowda, Krishnegowda Jayachandra.

**Methodology:** Gotravalli V. Rudresha.

**Project administration:** Rajesh Rajaiah, Bannikuppe S. Vishwanath.

**Supervision:** Rajesh Rajaiah, Bannikuppe S. Vishwanath.

**Validation:** Rajesh Rajaiah, Bannikuppe S. Vishwanath.

**Writing – original draft:** Gotravalli V. Rudresha, Rajesh Rajaiah, Bannikuppe S. Vishwanath.

**Writing – review & editing:** Gotravalli V. Rudresha, Amog P. Urs, Rajesh Rajaiah, Bannikuppe S. Vishwanath.

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
