## [Decision Letter · Decision Letter 0]

17 Aug 2020

Dear Dr. Vishwanath,

Thank you very much for submitting your manuscript "Echis carinatus snake venom metalloprotease-induced toxicities in mice: therapeutic intervention by a repurposed drug, tetraethylthiuram disulfide (disulfiram)" for consideration at PLOS Neglected Tropical Diseases. As with all papers reviewed by the journal, your manuscript was reviewed by members of the editorial board and by several independent reviewers. In light of the reviews (below this email), we would like to invite the resubmission of a significantly-revised version that takes into account the reviewers' comments. 

All reviewers agreed this study is interesting and presents a novel contribution to the growing body of research into potential antivenom-alternative therapies. However, all reviewers highlighted concerns with: a) clarity of methodology (particularly with in vivo experiments); b) inadequate discussion of limitations; c) insufficient context of the results and methods used with regards to real-life envenomation and similar studies assessing SVMP inhibitors. 

We cannot make any decision about publication until we have seen the revised manuscript and your response to the reviewers' comments. Your revised manuscript is also likely to be sent to reviewers for further evaluation.

Sincerely,

Stuart Robert Ainsworth

Associate Editor

Jean-Philippe Chippaux

Deputy Editor

Reviewer's Responses to Questions

**Key Review Criteria Required for Acceptance?**

**Methods**

-Are the objectives of the study clearly articulated with a clear testable hypothesis stated?

-Is the study design appropriate to address the stated objectives?

-Is the population clearly described and appropriate for the hypothesis being tested?

-Is the sample size sufficient to ensure adequate power to address the hypothesis being tested?

-Were correct statistical analysis used to support conclusions?

-Are there concerns about ethical or regulatory requirements being met?

Reviewer #1: The methods are generally clearly described with the following comments: 

1. The two methods most likely to result in questions refer to other papers and "modified appropriately". Since these involve quite graphic images, I think it is better to be explicit about how this was done and most importantly, how the animals were kept comfortable. For example, the animals were sedated for the Disulfiram paw injection, but what about the venom injection? for the intradermal and paw injection this is my advice, but I would leave it to the authors to decide how best to handle. 

2. "The signs and symptoms of toxicities were observed up to 24 h and survival time was

recorded. For systemic hemorrhage and bleeding studies, mice received ECV (LT50; 2.21 mg/kg,

i.p) and challenged with TTD (2.15 mg/kg; i.p) or ED ASV, 30 min post ECV injection. After 3 h

peritoneal cavity was observed for symptoms and photographed." 

Here I think you have substituted "symptoms" where you mean "signs". Symptoms are "headache, back pain, abdominal pain" (subjective) whereas signs are things like, "hemorrhage, weakness, guarding" (physically observable)

Reviewer #2: Are the objectives of the study clearly articulated with a clear testable hypothesis stated? Generally yes

Is the study design appropriate to address the stated objectives? Partially, see comments below.

Is the population clearly described and appropriate for the hypothesis being tested? Generally yes.

Is the sample size sufficient to ensure adequate power to address the hypothesis being tested? Yes

Were correct statistical analysis used to support conclusions? Partially, see comments below

Are there concerns about ethical or regulatory requirements being met? No concerns

The use of inhibitors of PLA2 and hyaluronidase activities is interesting and relevant for assessing the possible role of these other venom enzymes in the pathogenesis of tissue damage. However, the authors need to show (controls) that such inhibitors, at the concentrations used, indeed inhibited enzymatic PLA2 and hyaluronidase activities. These controls are necessary.

ECV-induced hemorrhage, line 5: Use ‘various doses of…’ instead of ‘various concentrations’.

Description of the necrosis model in footpad: The semiquantitative scale used for grading the extent of tissue damage includes the ‘necrosis’. From the text it seems that such effect was judged visually, i.e., macroscopically. It is difficult to identify necrosis without doing histological analysis. Was this done to corroborate whether the macroscopic observations agree with histological assessment? It is recommended that the macroscopic observations of necrosis be complemented by histological observations, at least with the crude venom alone.

The authors need to briefly explain why was DNAse used as positive control in this assay.

In the necrosis assay, the inhibition experiments described include a protocol in which the inhibitor was administered after venom injection. Did the authors also carried out experiments with preincubation of venom and inhibitors? The same holds for the inhibition of hemorrhagic activity. Were preincubation experiments also done? Preincubation type experiments are necessary to determine the inhibitor IC50 (median inhibitory concentration), which enables the comparison with other SVMP inhibitors previously published. It is recommended that the authors do these preincubation inhibition experiments, and determine the IC50.

Experiments on lethality, third line: LD50 instead of LT50. In the lethality experiments the authors used the preincubation protocol in addition to the ‘rescue’ protocol. Why was this not done with hemorrhage and footpad necrosis?

Statistics: A non-parametric test (Mann-Whitney U) was used to compare pairs of means, whereas a parametric test (ANOVA) was used for comparing means of more than two groups. If the authors decided that non-parametric tests were to be used, either because there is no evidence of a normal distribution of data or because the n is low, then both types of analyses should use non parametric tests and, therefore, Kruskal-Wallis test has to be used instead of ANOVA when comparing more than two groups.

Reviewer #3: (No Response)

**Results**

-Does the analysis presented match the analysis plan?

-Are the results clearly and completely presented?

-Are the figures (Tables, Images) of sufficient quality for clarity?

Reviewer #1: I have no question about the results. This is a very thorough investigation. It would be useful to have some clinical vision to illustrate how the authors might see this used and how the experimental doses compare to clinical observations. 

For example, in my experience, Echis bite patients don't die immediately. The inflammatory syndrome appears over days. Does this mean high dose IV in the hospital is a viable concept or should it be an oral field treatment? 

The authors admirably used rapidly lethal doses (hours). This speaks well to the concept. They should emphasize this. I doubt few people are envenomed with 3mg/kg venom. How would this drug be anticipated to perform at a lower dose, but more common bite presentation (for discussion eg). Would the dose of drug be lower and more plausible for pocket carriage?

Reviewer #2: Does the analysis presented match the analysis plan? Yes

Are the results clearly and completely presented? Generally yes.

Are the figures (Tables, Images) of sufficient quality for clarity? Yes.

Inhibition of gelatin hydrolysis by venom: As shown in Fig 1, not even a concentration of 20 mM of the inhibitor inhibited the activity. In the text, the authors claim ‘maximal inhibition was observed at 20 mM TTD”. The sentence is a bit confusing, because no complete inhibition was achieved and also because it is likely that higher inhibitor concentration would have a higher inhibitory activity. The inhibition of proteolytic activity of venom on various ECM substrates is more clear and well shown.

Inhibition of hemorrhage: As noted above, it is recommended that the authors carry out experiments with preincubation of venom and various concentration of the inhibitor in order to determine the IC50.

Reviewer #3: (No Response)

**Conclusions**

-Are the conclusions supported by the data presented?

-Are the limitations of analysis clearly described?

-Do the authors discuss how these data can be helpful to advance our understanding of the topic under study?

-Is public health relevance addressed?

Reviewer #1: As above, I think two areas of discussion are lacking and the authors might be short-changing themselves: 

1. What is the most common natural history presentation of Echis bite (rapid lethality or inflammatory/coagulopathy syndrome over period of days)? This puts clinical relevance and natural history into context and is very educational for readers, too. 

The citations used by the authors for general description of viper bites cites one basic research article (32) and then the other two describe a European viper and Russell's viper as examples. If Echis is so deadly...the reader would expect literature examples describing Echis. These references are fine, but I was a little surprised. There also seems to be regional variation in the morbidity and mortality from Echis (e.g. fatalities rare in Sri Lanka, even in Jaffna area). Mainly, I think it is important to understand about the NETs, how quickly the inflammatory effects typically manifest as systemic illness and VICC for example. The paper is very strong on the scientific underpinnings, but it is hard to connect this set of dots. I think the story told by these extensive experimental explorations will be enhanced by clinical context. There is no doubt Echis is extremely hard to manage both acutely and chronically. Where does this fit in? 

2. What are the best case and most likely clinical applications for this intervention?

Reviewer #2: Are the conclusions supported by the data presented? Generally yes, but see below.

Are the limitations of analysis clearly described? No, and on the basis of the comments indicated below, this will be necessary. The lack of histological assessment of necrosis is a limitation that needs to be discussed. My recommendation, however, is that the authors include this histological analysis to better describe the tissue necrosis.

Do the authors discuss how these data can be helpful to advance our understanding of the topic under study? Only partially. It is necessary to better place this study in the context of similar studies assessing the therapeutic potential of SVMP inhibitors.

Is public health relevance addressed? Yes, but it would be relevant to at least mention how would this inhibitor be administered in an eventual clinical use (route of injection).

There have been several studies exploring the use of synthetic inhibitors of SVMPs (including both chelating agents and peptidomimetic inhibitors) in experimental models of envenoming by various venoms. The authors need to mention these previous studies, in particular those carried out with Echis sp venoms. When doing this, it is relevant to compare the values of IC50s of some of these inhibitors vs disulfiram, because the concentrations of this inhibitor required to achieve inhibition of venom are in the mM range, while other inhibitors, including chelating agents are in the micromolar range. One wonders why the authors did not test for comparison other SVMP inhibitors previously tested which may be more potent than disulfiram and are commercially available. As mentioned above, it is necessary to find the value of IC50 of disulfiram for proteolytic and hemorrhagic activities through preincubation-type experiments.

The use of disulfiram has been associated with some adverse effects in patients. This may be mentioned in the Discussion.

Reviewer #3: (No Response)

**Editorial and Data Presentation Modifications?**

Reviewer #1: (No Response)

Reviewer #2: (No Response)

Reviewer #3: (No Response)

**Summary and General Comments**

Reviewer #1: Very nice paper. MRLewin

Reviewer #2: This is an interesting contribution that demonstrates that a chelating agent that inhibits SVMPs of Echis carinatus venom inhibits hemorrhagic, necrotizing, and lethal activities even when administered after venom injection, i.e. in rescue-type experiments. This underscores the relevance of SVMPs in the local and systemic effects induced by this venom. These findings are of value. The authors also evaluated the role of SVMPs in the generation of NETosis, which has been shown to be related to the pathogenesis of local necrosis. The possible role of PAR-1 and intracellular pathways in the genesis of NETosis by this venom is also shown. The main concerns I have with this work have been detailed in the comments to specific sections.

Reviewer #3: Rudresha et al. present a very interesting paper exploring the neutralisation of Echis carinatus venom (specifically the SVMP toxins) using a repurposed drug (disulfiram). The authors demonstrate interesting efficacy in a variety of animal models (haemorrhage, necrosis and lethality) and also explore the role of NETs in contributing to pathology. Overall, it’s a novel study, and the findings are really quite interesting. There are, however, a number of issues with the paper as it currently stands, though most relate to the text rather than the experimental approach (although see below). Currently, the rationale for various aspects of the work (inhibitor selection, experimental approach, etc) is unclear, with the reader left to piece it together themselves. Important information about the potential clinical utility of the disulfiram vs envenoming is also sadly lacking, including a critical evaluation of its likely utility. Finally, the animal experiments need to be much better defined – I found it impossible to robustly evaluate whether the venom challenges doses used in the lethality experiments were appropriate, for example, while details of the numbers of animals used in each experiment is missing. These are serious problems that currently undermine the manuscript. Despite these issues, there is potential for this study to be a novel and valuable contribution to the field, should the authors address the issues listed below. 

Major comments

The introduction would benefit from some text describing other studies where inhibiting SVMPs has resulted in promising preclinical efficacy vs snakebite envenoming. Reading on, the first 12 lines or so of the results incorporate text that would be useful in the introduction to help the reader understand the rationale better. Please move and integrate this from the results to the introduction. Some rationale for the use of TTD, AA and SLN would be really helpful for the reader – why might these be useful?! It’s not until the first page of the results that I read that TTD has chelating properties, and that AA and SLN are PLA2 and hyaluronidase inhibitors… The rationale for Zn chelation needs to be clearly articulated in the introduction somewhere. 

The scoring system used for the footpad experiments requires further detail given the subjective nature of these assessments. Given the nature of these experiments, were any ethical interventions used to reduce pain, suffering and distress of experimental animals? E.g. analgesia? Also, what was the rationale for SCH79797 before venom delivery, while TTD was given post-venom delivery? Why was the venom dose reduced for SCH79797 experiments? This needs to be explained. Finally, what does “LT50” stand for? This is not explained as far as I can see.

For the mortality studies, the above point about LT50 is important. Why was a known factor of the LD50 (e.g. 5 x LD50) not used? Details of the antivenom used need to be provided – including source, dose used and route of injection. Finally, for the hemorrhage study: why was the peritoneal cavity only observed in the 2 hour reduced venom challenge experiments, and not the 24 h experiments? Please also change “hemorrage and bleeding studies” to “hemorrhage study”, and “After 3 h peritoneal” to “after 3 h the animals were euthanized and the peritoneal cavity”

All animal experiments need to be clearly defined in terms of numbers of mice used per experimental arm. The details of n numbers also needs to be added to the figure legends. There are hints that the haemorrhage experiments are n=2, which is low, but its not detailed anywhere, so I cannot assess this. 

I don’t find the degradation gels for laminin (Figure 1, E) or Fibronectin (F) convincing. There are clear lower molecular weight degradation products in the laminin gel even at 20mM TTD concentration, and likewise, for the fibronectin gel the pattern does not match the control. There might be some inhibition, but its not particularly convincing. Also, Panel D in Figure 1 (collagen IV) needs repeating, because there is insufficient clarity in the image due to protein overloading. 

For the NET formation work, did the authors consider fractionating EC venom to obtain SVMPs to test directly to confirm their hypothesis that SVMPs are likely the causative agents (suggested by the TTD experiments)

In the discussion, the section about TTD requires some additional information. The text about ‘repurposing’ for cancer, AIDS and Lyme disease etc, needs an explanation of the type of studies cited – are the clinical trials? Preclinical studies? Or in vitro studies? Please detail. When discussing other metal chelators you should also cited the paper by Ainsworth et al. 2018 Communications Biology where EDTA was shown to protect against lethality caused by Echis envenoming in vivo. Why did the authors select TTD over other these SVMP inhibitors previously explored (and others, e.g. marimastat, batimastat)? An explanation of the rationale would be helpful. What are the potential drawbacks of TTD in terms of its potential translation for snakebite envenoming? Some discussion of its known PK, tolerance and route of administration should be discussed. What about the doses used in this study? How might they relate to doses required for clinical use? Given that there are a number of safety concerns with the use of TTD – these should be explicitly stated. The discussion would also benefit from the authors stating what needs to be done next to move this promising drug forward. 

Minor comments

Abstract: ECV definition needs explaining. 

Introduction: “never the less” should read “nevertheless”

Introduction: Echis should be italicised

Introduction: “Echis bite is responsible for the highest mortality” – I’m not sure there is any evidence for this in India. If there is, it should be cited. Also, “due to rich in zinc” should read “due to its venom being rich in zinc”

Introduction: “(NETs) the process is described as”, please edit to “(NETs) – this process is described as”

Introduction: “they showed that NETs” – who showed this?! Revise to “A previous study showed etc”. Also “lock” should be replaced by “entrap”

Introduction: “in this line we have used an Antabuse drug”. This doesn’t follow for the reader – you need to explain why disulfiram was tested based on the findings above. Also, replace “in this line” with “consequently” or something similar.

Methods: How long were the venoms stored at -20C for? 

Methods: please detail the sex of the mice used.

Methods: please add the conditions used for electrophoresis. 

Methods: Why were different doses of TTD, AA and SLN given to the mice when assessing haemorrhage? Why were these not consistent to enable comparison?

Methods: “analysis of NETs formationand” should read “Analysis of NET formation and”

Methods: please detail exactly what the lysis buffer containing EDTA free protease inhibitor mixture contains.

Methods: For western blotting, please provide the electro-blotting conditions used, the concentration of primary antibodies used, and what the TBST buffer contains.

Results: change “TTD also inhibited ECV-induced ECM proteins, collagen, laminin and fibronectin degradation” to “TTD also inhibited ECV-induced degradation of the ECM proteins, collagen, laminin and fibronectin, “

Results: change “skin hemorrhage in mice skin” to “murine skin hemorrhage”

Results: Sentence beginning “Recently, Katkar et al. reported” needs rephrasing.

Results: “ECV is lethal when injected at 3.31 mg/kg” – please provide the reference supporting this assertion.

Results: “therapeutic mode” might read better as “challenge then treat”

Results: “this indidcates that TTD is a potential drug candidate that complements ASV during EC bite” – reword. No evidence that this is a complementary treatment, instead used as a comparator. Please detail dosage of both TTD and ASV used so that the reader understands this clearly at this point in the manuscript. 

Results: For the neutralisation of lethality please describe the data. I.e. both TTD and ASV protected all experimental animals (n=5?) from the lethal effects of the venom in the 24 h coincubation model. For the challenge then treat model, four of five (?) experimental animals survived when treated with TTD etc.

Results: “Bothrops bilineata” should read “Bothrops bilineatus”

Results: sentence beginning “PARs are members” needs editing as currently it doesn’t make sense. Also “relatively MMPs are acted on” also needs editing.

Figure 3: I suggest that the authors crop the mouse images so that only the exposed peritoneal cavity is visible, and enlarge the images to allow better clarity. 

Discussion: “Viper bites predominately cause local tissue necrosis at the bitten site”. Actually this is not the most common sign of pathology – some victims suffer necrosis, but only a small percentage. Rephrase. 

Discussion: “Generally SVMPs contain..” – only the P-III SVMPs contains the D, C and M domains. The rest of the sentence about ADAMDEC-1 is unclear – this isn’t an SVMP. And again, later in the paragraph “the ADAMDEC-1 plays a crucial role in their enzymatic activities”. This is unclear – ADAMDEC-1 is not a domain of SVMPs…

Discussion: “chelation of Zn metal ion from SVMPs by specific Zn chelators rather than antivenom like molecule is more effective in the management…” Is it? What is the evidence for this?

PLOS authors have the option to publish the peer review history of their article (what does this mean?). If published, this will include your full peer review and any attached files.

Reviewer #1: Yes: Matthew R. Lewin

Reviewer #2: No

Reviewer #3: No
---

## [Decision Letter · Decision Letter 1]

13 Nov 2020

Dear Dr. Vishwanath,

Thank you very much for submitting your manuscript "Echis carinatus snake venom metalloprotease-induced toxicities in mice: therapeutic intervention by a repurposed drug, Tetraethylthiuram disulfide (Disulfiram)" for consideration at PLOS Neglected Tropical Diseases. As with all papers reviewed by the journal, your manuscript was reviewed by members of the editorial board and by several independent reviewers. In light of the reviews (below this email).

The scientific contribution of this study is substantial, and it is highly desirable that it be published. However, we are presently recommending a “major revision”. 

The reason for this is that both myself and Deputy editor are concerned regarding the ethics of the footpad model employed in this study. Moreover, all the reviewers, at one moment or another, have additionally questioned the ethical problem of this model. It is evident from the photographs in the figures that the mice would have experienced substantial pain and distress for an extended period. 

From the results presented in Fig 2 A/B/C, Fig 6 and Fig S3 D/E, it is difficult to understand the requirement to expand the experiment past day 4. Is the data obtained by day 4 not sufficient to demonstrate a clear conclusion? It is sufficient, then it is unnecessary to extend the suffering of the animals to day 8. 

We are therefore requesting further addition of discussion and clarification regarding this model. 

Specifically:

1) you need to unequivocally demonstrate, using strong arguments, that it is essential to prolong the monitoring of the mice in the footpad model for 8 days. 

Or

2) Demonstrate that you recognize that the elements obtained on the fourth day (or earlier) are sufficient to obtain a clear conclusion without difference to that of the eighth day, and therefore that it becomes unnecessary to prolong the suffering of the mice beyond this point. 

We recognize the urgent need for research investigating anti-necrotic snakebite therapies, and that due to the complex pathology of envenoming animal models are currently a necessity. However, it is essential to find a way to avoid the suffering of the mice during the entire duration of such experiments and not just at the time of the injection of the venom. If analgesia was used throughout the experiment, this has not been stated yet. Considerations of refining the protocol for future use (by yourselves or others) need to be included. 

We would like to invite the re-submission of a significantly-revised version that takes into account the above comments and reviewers' comments, below. 

We cannot make any decision about publication until we have seen the revised manuscript and your response to the reviewers' comments. Your revised manuscript is also likely to be sent to reviewers for further evaluation.

Sincerely,

Stuart Robert Ainsworth

Associate Editor

Jean-Philippe Chippaux

Deputy Editor

Reviewer's Responses to Questions

**Key Review Criteria Required for Acceptance?**

**Methods**

-Are the objectives of the study clearly articulated with a clear testable hypothesis stated?

-Is the study design appropriate to address the stated objectives?

-Is the population clearly described and appropriate for the hypothesis being tested?

-Is the sample size sufficient to ensure adequate power to address the hypothesis being tested?

-Were correct statistical analysis used to support conclusions?

-Are there concerns about ethical or regulatory requirements being met?

Reviewer #1: This is both impressive and done with sufficient detail that I feel as if I could readily reproduce or extend the studies described. I appreciate the diligent methodology and reporting.

Reviewer #2: The authors have adequately addressed my main concerns with the methods and have modified the paper accordingly. The inclusion of experiments with preincubation of venom and inhibitor was done.

Reviewer #3: (No Response)

**Results**

-Does the analysis presented match the analysis plan?

-Are the results clearly and completely presented?

-Are the figures (Tables, Images) of sufficient quality for clarity?

Reviewer #1: Yes. See below.

Reviewer #2: No major comments, except that I suggest the authors express the inhibitory activity on proteolysis and hemorrhage of the chelating agent in molar terms, i.e. expressing the Median Inhibitory Concentration (IC50) in molar terms. In the manuscript they present a graph but there is no reference to IC50. In order to compare this inhibitor with others, and to better express its inhibitory activity, IC50 should be expressed in molar terms in experiments where venom and inhibitors were preincubated. The authors already have the information to do so.

Reviewer #3: (No Response)

**Conclusions**

-Are the conclusions supported by the data presented?

-Are the limitations of analysis clearly described?

-Do the authors discuss how these data can be helpful to advance our understanding of the topic under study?

-Is public health relevance addressed?

Reviewer #1: Yes. See below.

Reviewer #2: The conclusions are sound and supported by the data.

Reviewer #3: (No Response)

**Editorial and Data Presentation Modifications?**

Reviewer #1: (No Response)

Reviewer #2: (No Response)

Reviewer #3: (No Response)

**Summary and General Comments**

Reviewer #1: This is both impressive and done with sufficient detail that I feel as if I could readily reproduce or extend the studies described. I appreciate the diligent methodology and reporting. 

I make two comments: 

Discussion: The LD50s (refs 50-52) do not match well with what you would expect if the effect of the venom was linear e.g. 6.5mg/kg and venom yield of 40mg...would kill 1/2 the people weighing six or seven Kg. What I think the authors intend from their work and other elements of the paper is to show that the effect of a small amount of venom produces an outsized effect on human victims through cascades of provoked inflammation. What is needed to produce lethality in mice is not necessary to make this argument coherent...mice and humans are not comparable for purposes of susceptibility to inflammation or the response to venom. Noting this does not in any way diminish the methods or overall conclusions. I would say it strengthens the arguments that pharmacological interventions that can neutralize venom and/or simply blunt the victims dysregulated response to venom or venom-induced tissue injury helps the case. Based on the reported LD50 (at least as stated), you would need much much bigger snakes to have serious lethality. A 40mg dose of venom is a relative under-dose. What is remarkable is how big an injury and degree of lethality that little amount of venom can produce. 

The vast majority of snakebites are from defensive strikes and minimal venom is injected. Yet, despite this massive "under-dosing" the bites produce devastating systemic effects (likely because of dysregulated host response--humans are not the intended victims of venom). Saw scale vipers are a particularly notable case. The patients rarely die quickly. The major damage occurs days or weeks after the bite. Understanding this is key to understanding the relevance of the work and for the authors to show why understanding the mechanisms--particularly for saw scale--is critical. A more explicit statement of the natural history of the bites would be helpful to put your findings in context for general readers. 

I think this reference is an omission and should be included (I am not any of these people): It probably belongs in the bibliography around reference 54 or so. 

Brust A, Sunagar K, Undheim EA, Vetter I, Yang DC, Casewell NR, Jackson TN, Koludarov I, Alewood PF, Hodgson WC, Lewis RJ, King GF, Antunes A, Hendrikx I, Fry BG. Differential evolution and neofunctionalization of snake venom metalloprotease domains. Mol Cell Proteomics. 2013 Mar;12(3):651-63.

Superb team effort to put this all together. 

Matthew R. Lewin, MD, PhD 

California Academy of Sciences

Reviewer #2: (No Response)

Reviewer #3: It’s nice to see that the authors have really engaged with the various reviewer suggestions and have integrated a number of changes into the manuscript. I am happy for the manuscript to be accepted, pending some minor corrections outlined below. The manuscript would also benefit from some English language improvements, which hopefully PLOS NTD will be able to facilitate. I congratulate the authors on a very interesting piece of work. 

Suggested minor changes:

Author summary: “Here we investigated the ECVMPs are responsible for”. ECVMP has not been defined at this point. 

Methods, in vivo experiments and elsewhere: “challenging” studies, would be better as “challenge studies”

Discussion, first line: Would read better as: “Viper bites can induce progressive tissue necrosis that can result in permanent disability in the affected limb or digit”

Discussion, second line: this sentence (beginning “there are several case reports”) needs rephrasing as it doesn’t currently make sense.

Discussion, lines 3-6: citations are needed for this information about Echis carinatus envenoming. 

Discussion, line 7: given the text about human envenoming above, the LD50 information needs to be clearer that this relates to animal studies! i.e. “in mice”

Discussion, line 11-12: my understanding is that Echis envenomings predominately act in a procoagulant manner by activating prothrombin, not Factor V or X.

My final suggestion would be that the authors write a final paragraph summarising the wider implications of their interesting findings and what should be done next to progress their work. Currently, the paper finishes quite abruptly. 

Accompanying raw data underpinning the manuscript should be made available in some form.

PLOS authors have the option to publish the peer review history of their article (what does this mean?). If published, this will include your full peer review and any attached files.

Reviewer #1: Yes: Matthew R Lewin

Reviewer #2: No

Reviewer #3: No
---

## [Editor Report · Decision Letter 2]

3 Jan 2021

Dear Dr. Vishwanath,

We are pleased to inform you that your manuscript 'Echis carinatus snake venom metalloprotease-induced toxicities in mice: therapeutic intervention by a repurposed drug, Tetraethylthiuram disulfide (Disulfiram)' has been provisionally accepted for publication in PLOS Neglected Tropical Diseases.

Best regards,

Stuart Robert Ainsworth

Associate Editor

Jean-Philippe Chippaux

Deputy Editor

---

## [Editor Report · Acceptance letter]

29 Jan 2021

Dear Dr. Vishwanath,

We are delighted to inform you that your manuscript, "* Echis carinatus  * snake venom metalloprotease-induced toxicities in mice: therapeutic intervention by a repurposed drug, Tetraethylthiuram disulfide (Disulfiram)," has been formally accepted for publication in PLOS Neglected Tropical Diseases.

Best regards,

Shaden Kamhawi

co-Editor-in-Chief

Paul Brindley

co-Editor-in-Chief
